**Bromine atom production and chain propagation during springtime Arctic ozone depletion**
**events in Barrow, Alaska**
Chelsea R. Thompson,[1,a,b] Paul B. Shepson,[1,2] Jin Liao,[3,c,d] L. Greg Huey,[3] Chris Cantrell[4,e],
Frank Flocke,[4] and John Orlando[4]
[1]Department of Chemistry, Purdue University, West Lafayette, IN, USA
[2]Department of Earth and Atmospheric Sciences and Purdue Climate Change Research Center,
Purdue University, West Lafayette, IN, USA
[3]School of Earth and Atmospheric Sciences, Georgia Institute of Technology, Atlanta, GA, USA
[4]National Center for Atmospheric Research, Boulder, CO, USA
[a]now at: Cooperative Institute for Research in Environmental Sciences, University of Colorado
Boulder, Boulder, CO, USA
[b]now at: Chemical Sciences Division, NOAA Earth System Research Laboratory, Boulder, CO,
USA
[c]now at: Atmospheric Chemistry and Dynamics Laboratory, NASA Goddard Space Flight Center,
Greenbelt, MD, USA
[d]now at: Universities Space Research Association, Columbia, MD, USA
[e]now at: Department of Atmospheric and Ocean Sciences, University of Colorado Boulder,
Boulder, CO, USA
*Correspondence to:* C. R. Thompson (chelsea.thompson@noaa.gov)
**Abstract.** Ozone depletion events (ODEs) in the Arctic are primarily controlled by a bromine
radical-catalyzed destruction mechanism that depends on the efficient production and recycling
of Br atoms.   Numerous laboratory and modeling studies have suggested the importance of
heterogeneous recycling of Br through HOBr reaction with bromide on saline surfaces. On the
other hand, the gas-phase regeneration of bromine atoms through BrO-BrO radical reactions has
been assumed to be an efficient, if not dominant, pathway for Br reformation and thus ozone
destruction.   Indeed, it has been estimated that the rate of ozone depletion is approximately equal
to twice the rate of the BrO self-reaction.   Here, we use a zero-dimensional, photochemical
model, largely constrained to observations of stable atmospheric species from the 2009 OASIS
campaign in Barrow, Alaska, to investigate gas-phase bromine radical propagation and recycling
mechanisms of bromine atoms for a seven-day period during late March.   This work is a
continuation of that presented in Thompson et al. (2015) and utilizes the same model construct.
Here, we use the gas-phase radical chain length as a metric for objectively quantifying the
efficiency of gas-phase recycling of bromine atoms. The gas-phase bromine chain length is
determined to be quite small, at <1.5, and highly dependent on ambient $O_3$ concentrations.
Furthermore, we find that Br atom production from photolysis of $Br_2$ and BrCl, which is
predominately emitted from snow and/or aerosol surfaces, can account for between $30 - 90\%$ of
total Br atom production.   This analysis suggests that condensed phase production of bromine is
at least as important as, and at times greater than, gas-phase recycling for the occurrence of
Arctic ODEs.   Therefore, the rate of the BrO self-reaction is not a sufficient estimate for the rate
of $O_3$ depletion.
**1  Introduction**

The springtime depletion of boundary layer ozone in the Arctic has been the subject of

intense research for several decades.   Early observations revealed a strong correlation between
ozone depletion events (ODEs) and enhancements in filterable bromine (Barrie et al., 1988).
This discovery led researchers to propose a mechanism for the bromine-catalyzed destruction of
ozone.
$Br_2 + hv \rightarrow 2Br$ (R1)
$Br + O_3 \rightarrow BrO + O_2$ (R2)
$BrO + BrO \rightarrow Br_2$ (or $Br + Br$) $+ O_2$ (R3)
This reaction cycle requires an initial source of bromine atoms to the boundary layer. Laboratory
and theoretical studies have suggested that $Br_2$ could be produced through oxidation of bromide
present in salt-enriched snow, ice or aerosol surfaces by gas-phase ozone (Hirokawa et al., 1998;
Oum et al., 1998b; Gladich et al., 2015).
$O_3 + 2Br^-_{(aq)} + 2H^+_{(aq)} \rightarrow\rightarrow Br_2 + O_2 + H_2O$ (R4)
Field observations by Pratt et al. (2013) using a controlled snow chamber experiment with
natural tundra snow collected near Barrow, AK lend further evidence to this mechanism, and
also suggest $Br_2$ production from OH produced photochemically within the snowpack. This
mechanism was further explored in the modeling study of Toyota et al. (2014) that suggested an
important role for this activation pathway in producing bromine within the snowpack interstitial
air.

Once present in the gas-phase, bromine atoms can be regenerated through radical-radical

reactions of BrO with XO (where X = Br, Cl, or I), NO, OH, or $CH_3OO$ to propagate the chain
reaction and continue the destruction cycle of ozone. If BrO photolyzes or reacts with NO, $O_3$ is
regenerated, and there is a null cycle with respect to $O_3$. However, although $O_3$ is not destroyed,
these two pathways represent efficient routes for Br atom propagation. Thus R3 serves to make
R2 effective in destruction of $O_3$. At the same time, Br atoms could be recycled through
heterogeneous reactions of HOBr with bromide in the condensed phase to release $Br_2$ to the gas-
phase via the now well-known "bromine explosion" mechanism (Vogt et al., 1996; Tang and
McConnell, 1996; Fan and Jacob, 1992).
$BrO + HO_2 \rightarrow HOBr + O_2$                   (R5)
$HOBr_{(g)} \rightarrow HOBr_{(aq)}$                     (R6)
$HOBr_{(aq)} + Br^-_{(aq)} + H^+_{(aq)} \leftrightarrow Br_{2(aq)} + H_2O$          (R7)
$Br_{2(aq)} \rightarrow Br_{2(g)}$                       (R8)
Evidence for reaction sequence R5 − R8 has been provided through laboratory studies, which
found that $Br_2$ was produced when frozen bromide solutions were exposed to gas-phase HOBr
(Huff and Abbatt, 2002; Adams et al., 2002). This mechanism is believed to proceed rapidly to
produce $Br_2$ so long as sufficient bromide is present in an accessible condensed phase. The
efficiency of this heterogeneous recycling mechanism has also been found to have a dependence
on the acidity of the surface, as was shown using natural environmental snow samples in Pratt et
al. (2013) and investigated in the modeling studies of Toyota et al. (2011, 2014), in a manner that
is consistent with the stoichiometry of Reaction R7.
To efficiently sustain the ozone destruction cycle to the point of near complete loss of
boundary layer ozone ($[O_3] < 2$ ppb), bromine atoms must be continually recycled through some
combination of the above mechanisms. The gas-phase reaction cycle described by Reactions R1
− R3 has generally been considered to be the dominant pathway for Br reformation following the
initial activation of $Br_2$ from the surface (the mechanism for which is still not fully understood).
Thus, it has been assumed that the rate of ozone destruction can be estimated as Equation 1 (see
Equation 15 in Hausmann and Platt, 1994, Equation 3 in Le Bras and Platt, 1995, and Equation 7
in Zeng et al., 2006), or as Equation 2 if chlorine chemistry is considered through Reaction R9
(Equation IX in Platt and Janssen, 1995).
$$-\frac{d[O_3]}{dt} = 2 \cdot k_3 \cdot [BrO]^2$$  (1)
$$-\frac{d[O_3]}{dt} = 2(k_3 \cdot [BrO]^2 + k_9 \cdot [BrO] \cdot [ClO])$$  (2)
BrO  +  ClO  →  Br  +  OClO  (R9)
However, these approximations assume that the ozone destruction rate is dominated by the BrO
+ XO reaction, which in turn necessitates efficient gas-phase recycling of Br; therefore, a
relatively long bromine chain length would be required to account for observed rates of ozone
destruction. It is, however, possible that Br atoms are generated mostly by $Br_2$ photolysis,
followed by BrO termination, e.g. by R5, in which case a short gas-phase bromine radical chain
length would be implied. The chain length for any process depends on the rates of the
propagation relative to the production and destruction reactions (Kuo, 1986). It is important to
note that, in our definition, the chain length refers to radical propagation reactions occurring
solely in the gas phase, and is a quantity completely independent of any condensed phase
chemistry. In the stratosphere, the Br/BrO catalytic cycle can have a chain length ranging from
$10^2$ to $10^4$ (Lary, 1996). In the troposphere, there is significantly less solar radiation and many
more available sinks; thus, radical chain lengths can be much shorter. For example, the chain
length of the tropospheric $HO_x$ cycle has been estimated to be ~ 4 − 5 (Ehhalt, 1999; Monks,
2005), increasing to 10 − 20 near the tropopause (Wennberg et al., 1998). The halogen radical
chain lengths in the Arctic troposphere have so far not been determined, thus, it is difficult to
evaluate whether Equations 1 and 2 are appropriate for estimating ozone depletion rates.
The importance of heterogeneous reactions for recycling reactive bromine has been
demonstrated in the recent literature (see review by Abbatt et al., 2012). Modeling studies using
typical Arctic springtime conditions to simulate ODEs have concluded that ozone depletion
cannot be sustained without considering the heterogeneous recycling of reactive bromine on
snow or aerosol surfaces (e.g., Michalowski et al., 2000; Piot and Von Glasow, 2008; Liao et al.,
2012; Toyota et al., 2014).  Michalowski et al. (2000) determined that the rate of ozone depletion
in their model was limited by the mass transfer rate of HOBr to the snowpack (effectively, the
rate at which Br is recycled through the heterogeneous mechanism) and that the depletion of
ozone is nearly completely shut down when snowpack interactions are removed.  Piot and von
Glasow (2008) simulated ozone depletion using the one-dimensional MISTRA model and
concluded that major ODEs (defined as complete destruction within 4 days) could only be
produced if recycling of deposited bromine on snow is included.  Without heterogeneous
recycling on the snowpack, the $BrO_x$ termination steps and irreversible loss of HOBr and HBr to
the surface prohibits the occurrence of an ODE.  More recently, using HOBr observations from
Barrow during OASIS, Liao et al. (2012b) found that a simple photochemical model over-
predicted observed HOBr during higher wind events ($> 6$ m s$^{-1}$), ostensibly due to an under-
predicted heterogeneous loss to aerosol in the model, and concluded that their field observations
support the hypothesis of efficient recycling back to reactive bromine via this mechanism.

It it is evident that the reactions occurring on snow and aerosol surfaces are likely the

initial source of halogen species to the polar boundary layer and that heterogeneous bromine
recycling on these surfaces must be considered for HOBr and HBr (as well as $BrNO_2$ and
$BrONO_2$ in higher $NO_x$ environments). However, the relative importance of gas-phase recycling
of bromine atoms is uncertain, even though is it often assumed that the ozone depletion rate can
be estimated reasonably well by the catalytic gas-phase radical reaction rates.  The goal of this
work was to investigate gas-phase Br atom propagation in terms of the bromine chain length in
comparison to the production of Br atoms through photolysis of $Br_2$ and BrCl, which are
predominantly produced directly from surface emissions and/or aerosol release. Here, we
present results from our study using a zero-dimensional model constrained with time-varying
measurements of molecular halogens, HOBr, $O_3$, CO, NO, $NO_2$, and VOCs from the 2009
Ocean-Atmosphere-Sea Ice-Snowpack (OASIS) campaign in Barrow, Alaska. This work builds
on the analysis presented in Thompson et al. (2015) using the same model framework. By
constraining our model with observations, we were able to conduct an in-depth study of the
halogen atom recycling occurring under varying conditions that were observed during the
campaign.

**2   Experimental**
**2.1   Measurements and Site Description**
The analysis presented herein utilizes observations conducted during the OASIS field
campaign that occurred during the months of February through April of 2009 in Barrow, AK.
The goal of the OASIS study was to investigate the chemical and physical processes occurring
within the surface boundary layer during ozone and mercury depletion events in polar spring.
This study resulted in the largest suite of simultaneous and co-located atmospheric measurements
conducted in the Arctic near-surface atmosphere to date, and represents a unique opportunity for
in-depth examination of a multitude of chemical interactions in this environment.
Atmospheric measurements were conducted from instrument trailers located near the
Barrow Arctic Research Consortium (BARC) facility on the Naval Arctic Research Laboratory
(NARL) campus. Winds arriving at the site are primarily northeasterly, from over the sea ice,
and thus represent background conditions with influence from natural processes and snow-air
interactions. Winds occasionally shift to westerly, bringing local emissions from the town of
Barrow to the site; however, these isolated events are easily identifiable by coincident
enhancements in both $NO_x$ and CO.

Measurements of molecular halogens, HOBr, NO, $NO_2$, $O_3$, CO, and VOCs were used to

constrain the model employed in this analysis. Instrumental methods for these measurements
have all been described elsewhere, thus, only a brief description is provided here. Inorganic
halogen species ($Br_2$, $Cl_2$, BrO, and HOBr) were measured by chemical ionization mass
spectrometry with $I^-$ ion chemistry as described in Liao et al. (2011, 2012, 2014); $O_3$, NO, and
$NO_2$ were measured by chemiluminscence (Ridley et al., 1992; Ryerson et al., 2000). CO was
measured using a standard commercial CO analyzer (Thermo Scientific) with infrared absorption
detection, and formaldehyde (HCHO) was measured at 1 Hz frequency using a tunable diode
laser absorption spectrometer, as described in Fried et al. (1997) and Lancaster et al. (2000). A
large suite of organic compounds was measured in situ by fast GC-MS (Apel et al. 2010) and via
whole air canister samples with offline GC-MS (Russo et al., 2010).

**2.2 Model Description**

The model used for this study is a zero-dimensional box model developed using the

commercial software FACSIMILE. A detailed description of the model can be found in
Thompson et al. (2015). We will describe the model only briefly here.

Our model consists of 220 gas-phase reactions and 42 photolysis reactions, representing

much of the known gas-phase chemistry occurring in the Arctic, including the important halogen,
$HO_x$, $NO_x$ and VOC chemistry associated with ozone depletions. The model also includes an
inorganic iodine reaction scheme adapted from McFiggans et al. (2000, 2002), Calvert and
Lindberg (2004) and Saiz-Lopez et al. (2008). Although IO has not been unambiguously
measured in the High Arctic above the ~1.5 – 2 pptv detection limit of LP-DOAS (long-path
differential optical absorption spectroscopy), observed enhancements in filterable iodide and
total iodine suggest that iodine chemistry is active to some extent in this region (Sturges and
Barrie, 1988; Martinez et al., 1999; Mahajan et al., 2010; Hönninger, 2002). Recently, $I_2$ has
been detected at tens of pptv within the snowpack interstitial air near Barrow, AK and at ≤0.5
pptv in the near surface air by I⁻ CIMS, providing direct evidence supporting the presence of at
least low levels of iodine chemistry (Raso et al., 2016). In our previous study (Thompson et al.,
2015), we investigated the impact of two different hypothetical levels of iodine. Here, we
investigate only the "Low Iodine" scenario for certain calculations, in which a diurnally varying
$I_2$ flux is incorporated such that average daytime mixing ratios of IO remain near 1 pptv for the
majority of the simulation. These levels of IO are realistic given our current knowledge based on
the work of Hönninger (2002) and Raso et al. (2016).
All gas-phase rate constants used in this model were calculated for a temperature of 248
K, consistent with average daytime conditions in Barrow for the period simulated. Although
some gas-phase reactions can exhibit a significant temperature dependence, we chose not to
incorporate variable temperatures into our model. This is justified in this case because ambient
temperature in Barrow for the week of 25 March 2009 varied by less than 10 K between the
maximum and minimum recorded daily temperatures. The radical oxidation and radical-radical
reactions that are of primary importance here do not have a large temperature dependence
(Atkinson et al., 2006, 2007); for example, a variability of 10 K imposes an ~1% change on the
rate of ethane oxidation by Cl atoms and a <4% change on the rate of the BrO + BrO radical self-
reaction. Most radical-radical reactions have only a small negative-temperature dependence.
Furthermore, and as mentioned previously, the major non-radical chemical species driving the
model are highly constrained to observations and are not allowed to freely evolve. Table 1
contains an abbreviated list of the reactions included in the model, showing only those reactions
that are central to the production, propagation, and termination of bromine radical chemistry,
which is the focus of this study.  A complete list of reactions can be found in Thompson et al.

(2015).

The model is configured to simulate 7 days during late March, 25 through 31 March, that

include a period of depleting ozone, a full ozone depletion ($[O_3] < 2$ ppbv) lasting for ~ 3 days,
and recovery.  The $O_3$ time-series for this period is shown in Figure 1A, along with radiation as a
reference (all plots are in Alaska Standard Time).   We constrain the model to observations for
this time period by reading in time-varying data sets of $O_3$, $C_2H_2$, $C_2H_4$, $C_2H_6$, $C_3H_8$, $C_3H_6$, $n$-
$C_4H_{10}$, $i$-$C_4H_{10}$, HCHO, $CH_3CHO$, $CH_3COCH_3$, methyl ethyl ketone (MEK), $Cl_2$, $Br_2$, HOBr, NO,
$NO_2$, and CO at ten-minute time steps.  All other gas-phase species are allowed to freely evolve.
Surface fluxes (represented as volumetric fluxes) are used for HONO and $I_2$ and are scaled to
$J(NO_2)$ as a proxy for radiation as both of these species are likely to be produced
photochemically.  Further discussion regarding HONO can be found in Thompson et al. (2015).

Photolysis rate constants ($J$ coefficients) for many of the species included were calculated

during OASIS using the Tropospheric Ultraviolent and Visible Radiation model from
measurements of down-welling actinic flux conducted throughout the campaign (Shetter and
Müller, 1999; Stephens et al., 2012).  Estimates of $J_{max}$ in the Arctic for OClO were taken from
Pöhler et al. (2010), for HOCl from Lehrer et al. (2004), and for $CHBr_3$ from Papanastasiou et al.
(2014). $J_{max}$ values for the iodine compounds were calculated according the work of Calvert and
Lindberg (2004), which also simulated conditions for late March in Barrow, Alaska.   Time-
varying $J$ coefficients for $O_3$ and $NO_2$ were read into the model at 10-minute time steps. All other
photolysis reactions were scaled to $J(NO_2)$ in the modeling code using the maximum $J$
coefficients ($J_{max}$) for 25 March (a clear-sky day) as a basis for scaling.  For cloudy days, this
method assumes that $J$ coefficients for other photolytically-active species are attenuated in a
manner that is proportional to $J(NO_2)$.

In the initial development of the model, heterogeneous reactions of halogen species

occurring on aerosol and snowpack surfaces were included, as well as mass transfer and dry
deposition for certain species using the method and mechanism of Michalowksi et al. (2000).
This mechanism assumes aqueous phase kinetics for those reactions occurring within a
uniformly distributed quasi-liquid layer (QLL), in a similar fashion as numerous other models
(e.g., Piot and von Glasow, 2008; Thomas et al, 2011; Toyota et al., 2014).  It was originally
intended to utilize this multiphase chemistry to produce halogen radical precursors.  However,
the heterogeneous production mechanisms could not reproduce observed $Br_2$ or $Cl_2$ from OASIS.
This reflects the complex but not fully understood condensed phase chemistry and physics that
leads to production of $Br_2$ (and $Cl_2$) (Abbatt et al., 2012; Domine et al., 2013; Pratt et al., 2013).
Additionally, the current knowledge of the physical properties of the QLL and the location of
liquid-like surfaces on snow grains would seem to invalidate the aforementioned assumptions on
which many of the current heterogeneous models are based (Domine et al., 2013), specifically
that the chemistry occurs in a liquid-like environment on snow grains. Indeed, Cao et al. (2014),
adopted a simplified heterogeneous chemistry mechanism in their modeling of Arctic ozone
depletion, wherein they use the mass transfer of HOBr to the surface as the rate-limiting step in
$Br_2$ production, citing the lack of suitable reaction mechanisms with which to properly simulate
condensed phase chemistry on snow/ice.  Admittedly, our model is also not able to capture these
complex heterogeneous processes. However, as discussed thoroughly by Domine et al. (2013),
even our most complex state-of-the-art snow chemistry models are neither physically nor
chemically accurate, and rely upon a variety of adjustable parameters to produce reasonable
results, because of the lack of fundamental understanding of the actual phase and physical and
chemical environment in which the chemistry is occurring.  It is thus clear that the kinetics of the
individual reactions in such a case cannot be reliably simulated.

In light of the limitations of all models of cryosphere photochemistry, a strength of this

study, and opportunity, rests with the fact that we have observations of key halogen species,
including $Br_2$, $Cl_2$, BrO, ClO, HOBr, as well as VOCs, $NO_x$, OH and $HO_2$.  Therefore, to study
the gas phase recycling discussed in the Introduction, in this work $Br_2$ and $Cl_2$ concentrations
were fixed at the observed levels (see Thompson et al., 2015 for further discussion) and were not
produced via heterogeneous chemistry.  During a period spanning a portion of 29 and 30 March,
$Br_2$ observations are not available due to instrument instability. Here, we have filled in the
missing portion of data with average daytime $Br_2$ values based on observations from 27 and 28
March and the morning data available for 29 March, and use average nighttime values for the
night of 29/30 March using the observations from the two adjacent nighttime periods. The filled-
in values for $Br_2$ result in reasonable agreement between modeled and observed BrO for this
period. In the analyses presented in Figures 3 and 5 – 10 we have indicated this period of missing
and filled-in $Br_2$ values with a shaded box. Due to the sparseness of BrCl observations during
OASIS, only daytime BrCl was used as produced in the model multiphase mechanism.  While
we do not argue that the production mechanism for BrCl is accurate, the daytime simulated BrCl
mixing ratios of 0 – 10 pptv are in approximate agreement with the available data for the
campaign.  In any case, according to our model, BrCl was not a significant source of either Br or
Cl atoms relative to $Br_2$ and $Cl_2$.

Though we do not use the heterogeneous chemistry module for any chemical production

(other than BrCl), deposition and mass transfer is a significant and critical sink for certain
species. Thus, we do make use of this aspect of the multiphase portion of the model, as described
below. The dry deposition velocity of $O_3$ to the snowpack is estimated at 0.05 cm·s$^{-1}$, consistent
with previous measurements and modeling studies (Gong et al., 1997; Michalowski et al., 2000;
Helmig et al., 2007; Cavender et al., 2008), though it is recognized that there is large uncertainty
with this parameter from field observations (Helmig et al., 2007, 2012).  Assuming a boundary
layer height of 300 m, this corresponds to a transfer coefficient, $k_t$, of 1.67x10$^{-6}$ s$^{-1}$.  Though we
have incorporated the deposition of $O_3$ in the model, the *gas-phase* mixing ratio of $O_3$ is
constrained to observations, which adjusts on 10-minute time steps. Dry deposition velocities for
the stable Arctic environment have not been determined for the halogen acids (HBr, HCl, HOBr,
HOCl, HOI), therefore we use the estimation method of Michalowski et al. (2000) and assume a
deposition velocity that is 10 times greater than for $O_3$, leading to a $k_t$ of 1.67x10$^{-5}$ s$^{-1}$.  In most
model runs, we have chosen to constrain to HOBr observations (further described in Section 3.1),
and thus a similar situation exists as for $O_3$ mentioned above. We assume an equivalent $k_t$ for the
oxidized nitrogen compounds (HNO$_3$, HO$_2$NO$_2$, HONO, N$_2$O$_5$, BrNO$_2$, and BrONO$_2$). The mass
transfer coefficient of atmospheric species to the particle phase is calculated as a first-order
process as described in Jacob (2000). The concentration of aerosol surface area used was 3.95 x
10$^{-7}$ cm$^2$ cm$^{-3}$ as calculated by Michalowski et al. (2000) from measurements made at Alert by
Staebler et al. (1994), with a maximum aerosol radius of r = 0.1 μm.  These levels are also
consistent with observations of aerosol surface area at Barrow, which ranged between 9 x 10$^{-8}$
cm$^2$ cm$^{-3}$ and 40 x 10$^{-7}$ cm$^2$ cm$^{-3}$ (Liao et al. 2012b). We recognize that this constant level of
aerosols imparts a constant loss rate in the model and does not take into account any variability
in the uptake strength. Because many of these species are lacking empirically-derived deposition
velocities (e.g, HOBr), there is necessarily large uncertainty in these values, and it is not possible
at this time to ascertain whether the uncertainty associated with the deposition velocity
estimation is greater or less than the uncertainty imposed by using a constant aerosol surface area.
Liao et al. (2012b) did use time-varying aerosol surface area from observations at Barrow,
however, they suggested that simple parameterization of deposition of HOBr to aerosols was
insufficient for accurately simulating HOBr (further discussion of HOBr is in Section 3.1). Given
the highly simplified nature of the surface deposition in our model, we do not attempt to
differentiate between aerosol uptake and deposition to the snow surface, and instead we lump
these two terms together under the "surface deposition" umbrella.  However, while we mostly
constrain the model to observed HOBr, the comparison to simulated HOBr using these values is
instructive.

**3   Results and Discussion**
**3.1   Comparison of modeled and observed Br$_2$, BrO, HOBr, and HO$_2$**

This work focuses on the propagation and production mechanisms of Br atoms, and thus

it is critical that our model accurately captures BrO and Br$_2$ at mixing ratios that are consistent
with observations.  Figures 1B and 1C show comparisons between modeled mixing ratios (black
trace) of Br$_2$ and BrO compared to the measured values during this time (red data) by chemical
ionization mass spectrometry (CIMS) (Liao et al., 2012b).  Modeled BrO is presented as hourly
averages. In the model, Br$_2$ is fixed to time-varying observations, whereas BrO is produced
strictly through the gas-phase photochemical reactions.  The model captures the overall temporal
profile and magnitude of BrO throughout the period. It should be noted, however, that the
uncertainty in the BrO measurements is large during ODEs as the observed values are very near
the detection limit (LOD of ~2 pptv with uncertainty of -3/+1 pptv near the LOD).

$Br_2$ mixing ratios reach 2 – 12 pptv (Figure 1B) during the daytime. Given the short

lifetime of $Br_2$ resulting from rapid photolysis, these daytime mixing ratios imply a large surface
flux, that in turn produces the BrO mixing ratios observed.  These $Br_2$ levels are consistent with
previous Arctic measurements that observed daytime $Br_2$ up to 27 pptv (Foster et al., 2001) and
agree well with the "uncorrected" $Br_2$ data reported in Liao et al. (2012a, 2012b) for this period.
It has been suggested that daytime $Br_2$ greater than the CIMS instrumental detection limit (~1
pptv) is an artifact of HOBr conversion to $Br_2$ on the instrument using an aircraft inlet (Neuman
et al., 2010), however, for the instrument configuration employed during OASIS, it is not clear
how much, if any, of the $Br_2$ signal is a result of HOBr reactions on instrument surfaces.

An estimate of the effective mixing height of $Br_2$ can be calculated using the method of

Guimbaud et al. (2002) and using an average measured diffusivity during OASIS of 1500 $cm^2$ $s^{-1}$
(R. Staebler, personal communication, 2015). By assuming that photolysis is the dominant loss
mechanism controlling the $Br_2$ mid-day lifetime in a stable boundary layer typical of Arctic
conditions, the daytime effective mixing height is ~1.85 m. This also assumes that the snowpack
is the primary source of $Br_2$ emissions, which is consistent with previous assumptions for the
aldehydes (Sumner et al., 1999; Guimbaud et al., 2002) and is supported by direct empirical
evidence of the tundra snowpack being a relatively strong source of $Br_2$ (Pratt et al., 2013).
Enhanced $Br_2$ within the snowpack interstitial air has also been predicted from the modeling
studies of Toyota et al. (2011, 2014). From this estimation, a significant fraction of the $Br_2$

present at the surface would remain at the height of the instrument inlet (~1 m) in the sunlit periods. If aerosols do represent a significant source of $Br_2$ as has been hypothesized, and inferred indirectly from bromide depletion in sea salt aerosols (Sander et al., 2003), then one would expect enhanced $Br_2$ to be present throughout the height of the boundary layer. In our highly constrained model, daytime $Br_2$ mixing ratios greater than 1 pptv are necessary to reproduce observed BrO, therefore, this modeling study suggests that $Br_2$ should indeed be present and above the instrument detection limit during the daytime. Br atoms are predicted at concentrations ranging from $1 \times 10^7$ to $3 \times 10^9$ molecules $cm^{-3}$. The hourly averaged model output for Br is shown in Figure 2D. No direct measurements of Br atoms are available with which to compare, though these values are within the range of estimates determined by Jobson et al. (1994) and Ariya et al. (1998).

In the case of HOBr, our model originally simulated this species based on the known gas-phase sources and sinks (including photolysis) and deposition/uptake to surfaces as described above. As shown in Thompson et al. (2015), and again in Figure 2A, given the observed $Br_2$ mixing ratios, the model greatly overestimated HOBr. Liao et al. (2012b) simulated inorganic bromine species from the OASIS campaign using a simple steady-state model and experienced that their model also overestimated the observed HOBr, with the overestimation becoming especially pronounced during periods of higher winds. They suggested a faster heterogeneous loss to aerosols or blowing snow that was not represented in their model, despite utilizing time-varying aerosol surface area from observations. For the majority of the results presented in this work, we chose to operate our model constrained to HOBr observations, as illustrated in Figure 2B. Figure 2C shows modeled HOBr obtained by adjusting the deposition to aerosols based on daily wind speeds (resulting in $k_t$ values ranging from $1 \times 10^{-2}$ to $1.5 \times 10^{-4}$ $s^{-1}$), and tuned to

provide reasonable agreement with observations. This resulted in smaller deposition rates on 25
through 27 March when winds were calm, and higher deposition rates on 29 through 31 March
when winds were up to 9 m/s. This method allowed us to calculate the importance of surface
deposition of HOBr relative to photolysis as a sink for this compound, but the constrained
version of the model was used for all other calculations, e.g. for the chain length calculations.
$HO_2$ is essential for the heterogeneous recycling of bromine (via Reactions R5 − R7).
Therefore, it is important that our model provides a reasonable estimation of $HO_2$ for this
analysis. In Figure 1E we show a comparison of simulated, hourly-averaged $HO_2$ (black trace)
and observed $HO_2$ from OASIS for this period (red data), measured using a CIMS developed for
peroxy radicals (Edwards et al., 2003). The range of daytime $HO_2$ mixing ratios is reproduced
reasonably well. Simulated $HO_2$ is on the lower limit of observations for 25 and 29 March, and
does not reach the maximum mixing ratios observed. The model also somewhat overpredicts
$HO_2$ on 28 through 30 March; however, the model values are within the stated 25% - 100%
range of uncertainty of the measurement.

**3.2 Chain length**
The ozone destruction cycle as described in Reactions R1 − R3 is a chain reaction
mechanism catalyzed by $BrO_x$. The effectiveness of a catalytic cycle can be quantified by
considering the chain length, that is, the number of free radical propagation cycles per
termination or per initiation. The radical chain length is a metric that refers solely to gas phase
reactions (Monks, 2005). We have not, until the OASIS2009 campaign, had the high quality
measurements available to enable a reliable estimation of the bromine radical chain length in the
Arctic.
The length of the chain in a radical propagation cycle is limited by termination steps that
destroy the chain carriers and result in relatively stable atmospheric species.  Thus, the chain
length can be defined as the rate of propagation divided by the rate of termination.  Alternatively,
the chain length can also be calculated using the rate of initiation.  If the total bromine radical
population is at steady-state, the rate of initiation is equal to the rate of termination; thus, for
short-lived radical species, the two methods for calculating chain length should be approximately
equal.
Method 1:     $\Phi = \frac{\Sigma(\text{Rates of propagation})}{\Sigma(\text{Rates of termination})}$     (3)
Method 2:     $\Phi = \frac{\Sigma(\text{Rates of propagation})}{\Sigma(\text{Rates of initiation})}$     (4)
We used our model to calculate the chain length for bromine radical propagation across
the 7-days of the simulated period using both Method 1 and 2 as shown in Equations 5 and 6.
Because bromine radicals are generated photolytically, the chain length is calculated for daytime
only, defined here as approximately 7:00 to 20:00 Alaska Standard Time (AKST).

Method 1:     $\Phi_{Br} = \frac{\begin{aligned} &(2k[BrO]^2 + J_{BrO}[BrO] + k[BrO][ClO] + \\ &k[BrO][IO] + k[BrO][CH_3OO] + \\ &k[BrO][OH] + k[BrO][O(^3P)] \\ &+ k[BrO][CH_3COOO] + k[BrO][NO]) \end{aligned}}{\begin{aligned} &(k[Br][HO_2] + k[Br][C_2H_2] + k[Br][C_2H_4] \\ &+ k[Br][C_3H_6] + k[Br][HCHO] + k[Br][NO_2] \\ &+ k[Br][CH_3CHO] + k[Br][C_3H_6O] + k[Br][C_4H_8O] \\ &+ k[Br][CH_3OOH] + k[BrO][HO_2] + k[BrO][CH_3OO] \\ &+ k[BrO][C_3H_6O] + k[BrO][NO_2]) \end{aligned}}$     (5)

Method 2:     $\Phi_{Br} = \dfrac{(2k[BrO]^2 + J_{BrO}[BrO] + k[BrO][ClO] + k[BrO][IO] + k[BrO][CH_3OO] + k[BrO][OH] + k[BrO][O(^3P)] + k[BrO][CH_3COOO] + k[BrO][NO])}{}$     (6)

$(2J_{Br2}$ [Br$_2$] + $J_{BrCl}$[BrCl] + $J_{HOBr}$[HOBr] + $J_{BrONO2}$[BrONO$_2$]

+ $J_{IBr}$[IBr] + $J_{BrNO2}$[BrNO$_2$] + $J_{CHBR3}$[CHBr$_3$] +

$k$[HBr][OH] + $k$[CH$_3$Br][OH] + $k$[CHBr$_3$][OH])


Termination reactions for bromine include those reactions that are sinks for either Br and BrO,
since Br and BrO rapidly interconvert. Here, photolysis of BrO and the BrO + NO reaction is
included in the numerator because they are efficient at reforming Br and propagating the chain;
however, these reactions do not result in a net loss of ozone. Photolysis of BrO produces atomic
oxygen that reacts with O$_2$ to form O$_3$, and NO$_2$ can photolyze to similarly reform O$_3$. Therefore,
it should be noted that if we omit these reactions and consider only those that result in a net O$_3$
loss, it would be expected that the chain length would be shorter. Indeed, model simulations were
performed without these two terms and the determined chain lengths were on average 80% lower
than those presented here. BrO reaction with CH$_3$OO is included in both the numerator and
denominator in Equation 5 because this reaction has two channels, one that propagates the Br
chain and one that terminates it.

In Figure 3, we present the hourly-averaged results of these calculations for the Base

Model, which show that the two methods for calculating bromine chain length are in reasonably
good agreement, although there are small differences between the two methods throughout the
time-series. This agreement is a test of our basic understanding of the radical chemistry. The
inset graph in Figure 3 shows a linear regression of the two methods for the chain length
calculation. The coefficient of determination ($r^2$) of 0.93 confirms the good temporal agreement
between the two methods. However, the slope of 0.68 indicates that Method 1 is generally higher
than Method 2 throughout (with some periods of exception). This offset reveals that either
Method 1 is slightly overestimating the chain length, or that Method 2 is underestimating it. The
numerator is identical in Equations 5 and 6, therefore, the denominator must be driving this
discrepancy, with either the denominator term in Method 1 too low or the denominator term in
Method 2 too high (or some combination thereof). If it's the case that the Method 1 denominator
is too low, then it must be concluded that there are important $BrO_x$ terminations that are missing
from the calculation. If, however, the denominator of Method 2 is too high, this would imply that
our measurements of these $BrO_x$ precursors are too high, which, as discussed above, is a known
possibility at least for the $Br_2$ measurements. The photolysis of $Br_2$ is the dominant initiation
pathway (see Section 3.3), therefore, the Method 2 chain length calculation would be the most
sensitive to $Br_2$ measurement inaccuracies.

In Equation 6, we included photolysis of the most prevalent organobromine compound

bromoform for completeness, though it has been recognized for many years that the rate of Br
atom production from this pathway is small (e.g., ~ 100 molecules·cm$^{-3}$·s$^{-1}$ for bromoform at
mid-day) compared to Br atom production from $Br_2$ photolysis (~1.3x10$^7$ molecules·cm$^{-3}$·s$^{-1}$ at
mid-day assuming 5 pptv of $Br_2$). Photolysis of bromine nitrate ($BrONO_2$) and nitryl bromide
($BrNO_2$) are also included, however, the prevalence of and production of these compounds in the
Arctic is highly uncertain, and no observations of these species in the Arctic have been published
to date with which to compare to our modeled mixing ratios. Inclusion of these terms at the
modeled $BrONO_2$ and $BrNO_2$ mixing ratios has a small effect on the calculated chain length that
cannot account for the discrepancy between the two methods.

The median bromine chain-length in the Base simulation, averaging the results from

Method 1 and Method 2, is ~1.2 across daylight hours (7:00 to 21:00 AKST) and ~2 for
afternoon hours, defined for this purpose as approximately 12:00 until 18:00 AKST, when $[O_3] \geq$
5 ppbv. In comparison, the bromine chain length is ~0.4 when $[O_3] < 5$ ppbv (Figure 3). In other
words, the chain cannot be maintained when $[O_3] < 5$ ppbv. Under these conditions, Br atoms
readily terminate, e.g. via reaction with $CH_3CHO$ (see below). On 29 March there is an early
morning enhancement in the chain length. This morning spike appears to correlate with a similar
sharp increase in ozone. $Br_2$ accumulates during the nighttime hours, resulting in the highest $Br_2$
concentrations in the early morning hours (Figure 1B). When the sun rises, $Br_2$ photolyzes
rapidly, releasing a pulse of reactive bromine that converts to BrO in the presence of ozone. This,
in concert with the coincident increase in ozone, can explain the enhanced chain length during
the early morning hours.

Overall, midday bromine chain lengths remain near or below 2 during background $O_3$

days. This implies that, for these days, ozone depletion is strongly dependent upon initiation
processes, and most BrO radicals produced terminate the chain via reactions R5 and R10 (see
below) in less than two cycles. Reaction R12 (see below) will also efficiently terminate the chain,
however, the relative importance of R10 and R12 depend upon the relative abundances of BrO
and Br. For background $O_3$ days, such as 29 and 30 March, [BrO] > [Br], thus, R10 > R12. The
low chain lengths calculated here are surprising, given that it has been generally accepted that Br
is recycled efficiently in the gas-phase. That it appears this is not the case supports the
conclusions of Michalowski et al. (2000), Piot and von Glasow (2008), and Toyota et al. (2014)
that heterogeneous recycling through the "bromine explosion", which emits $Br_2$ and BrCl from
surface reactions, is of critical importance to sustain ODEs occurring at the surface.

A question to address regarding the relatively small chain length calculated for Br is to

what extent the chain length is dependent on $NO_2$. As discussed in Thompson et al. (2015) and
further investigated in Custard et al. (2015), $NO_2$ at Barrow can be greater and more variable
than at very remote sites due to its proximity to anthropogenic emissions sources. We find that
the chain length calculation is relatively insensitive to $NO_2$ concentrations and so it is robust for
the range of conditions encountered at Barrow. This is shown in detail in Custard et al. (2015).
As discussed by Custard and coworkers, while $NO_2$ can inhibit the bromine chain through
reactions R10 and R12 (i.e., decreasing the chain length), enhanced $NO_2$ will also reduce
available $HO_2$, thereby decreasing the $HO_2$ available to terminate the chain (i.e., increasing the
chain length). While the Method 2 calculation does not contain $NO_2$ in the denominator, the
absolute [BrO] is $NO_x$-dependent because of reaction R10 (Custard et al., 2015), and it is through
this effect that high $NO_x$ mixing ratios act to decrease the rate of $O_3$ depletion. In the natural
environment, $Br_2$ production can potentially also be $NO_x$-dependent, e.g. via reaction R11,
followed by R7. While our model does not *simulate* the condensed phase processes, it is
implicitly sensitive to them, since the model is constrained to the product of those processes, $Br_2$.

$BrO$   +      $NO_2$  →      $BrONO_2$             (R10)

$BrONO_{2(aq)}$ + $H_2O$  →      $HOBr$ + $HNO_3$        (R11)

$Br$    +      $NO_2$  →      $BrNO_2$              (R12)

On the other hand, for the period of 26 through 30 March, $NO_x$ was relatively low, and the
relatively good agreement between the two calculation methods further supports our conclusion.

To investigate how chemical interactions with chlorine and iodine affect the bromine

chain length, a series of simulations was performed by varying the combinations of halogens
present in the model. The bromine chain length was determined for scenarios with only Br, Br
and Cl (Base Model), Br and Iodine, and Base with Iodine. Simulations without chlorine were
performed simply by removing $Cl_2$, while simulations with iodine were performed by
incorporating the $I_2$ flux as described in Section 2.2. No other adjustments were made to the
model for these sensitivity runs.
Table 2 shows the results for both chain length calculation methods (i.e., Equations 5 and
6) for the different halogen combinations for the three days when ozone was present near
background values: 25, 29 and 30 March. For the Base scenario ("Br and Cl"), the average of the
median daily values for the bromine chain length is 1.43 and 1.05 for Method 1 and Method 2,
respectively.  In comparison with the "Br Only" run, Cl chemistry does not induce a net increase
in the Br chain length, but rather causes a slight decrease.  Cl chemistry can increase Br radical
propagation through the addition of the BrO + ClO cross-reaction and enhancement of the BrO +
$CH_3OO$ radical propagation terms.  However, Cl chemistry can also increase the concentration of
reactive bromine sinks, such as aldehydes (e.g., propanal and butanal, which were free to evolve
in our model; HCHO and $CH_3CHO$ are fixed to observations) and $HO_2$ (see Thompson et al.,
2015). Iodine has the effect of increasing the Br chain length. When low levels of iodine are
added to the "Br Only" simulation, the chain increases from 1.52 to 1.59 in the Method 1
calculation, primarily due to the very fast cross-reaction between IO and BrO.  The addition of
Cl to the "Br and I" simulation imparts a slight decrease to the Br chain length.  This may be
explained by the competition between BrO and ClO for reaction with NO and/or IO, as well as
the additional Br sinks in the presence of Cl chemistry.  Regardless, overall there is more Br
available for reaction with $O_3$ when Cl is present due to the interhalogen reactions, thereby
increasing the rate of ozone depletion (see Thompson et al., 2015 for further discussion on ozone
depletion rates).
There are several conclusions that can be drawn from Figure 3 and Table 2: 1) there is a
distinct difference in bromine chain length between $O_3$-depleted and non-depleted days with a
significantly larger chain length when ozone is present, and 2) for all simulations, the average
bromine chain is much shorter than often expected (given that gas-phase recycling has, to date,
been assumed to be highly efficient).  The chain length is greatest when ozone is present because
many of the species that propagate the Br chain (e.g., BrO, ClO, IO, and to a lesser extent OH
and $CH_3OO$) require $O_3$ for production.  Although the relationship between bromine chain length
and BrO is not straightforward due to the multitude of interactions between BrO and other
species that either propagate or terminate the chain, the chain length does exhibit a rough
dependence on [BrO], as shown in Figure 4, that can be loosely described with a linear fit.  If it
were the case that the gas-phase Br chain length was relatively long (such that the numerator far
outweighs the denominator), and dominated by the BrO self-reaction, the numerator in Equations
5 and 6 would reduce to $2k[\text{BrO}]^2$, and the regression in Figure 4 would display a quadratic fit;
however, that is not observed here.

For purposes of comparison, the chain lengths for Cl and I were also calculated in a

manner analogous to that of Equation 5. These results are shown as hourly averages in Figure 5
for the Base with Iodine scenario.  It is apparent from this figure that reactive chlorine exhibits
an exceptionally short chain length, whereas reactive iodine has a relatively long chain length.
The average Cl chain length across the three days of background ozone (25, 29, and 30 March) is
0.15, or 0.23 considering only afternoon hours (12:00 – 18:00 AKST). This result indicates that
nearly all Cl atoms that are produced terminate, likely through the very efficient reaction with a
multitude of VOCs, as shown in Thompson et al. (2015).  This behavior also helps explain why
Cl has only a small effect on the bromine chain length.  In contrast, I and IO have few known
sinks, which results in a reactive iodine chain length of 5.7 on average over 25, 29, and 30 March,
and 7.3 over only mid-day hours, with maxima over 12. The high efficiency of the gas-phase
regeneration of I in part explains why iodine is more efficient on a per atom basis at depleting
ozone than either Br or Cl (Thompson et al., 2015).

## 3.3  Reactive bromine initiation, propagation, and termination pathways

The individual reactions that initiate, propagate, and terminate the reactive bromine chain

were examined to determine the most important reaction pathways contributing to the chain
reactions.   The rates of Br atom production from the most important initiation pathways are
shown as hourly averages in Figure 6, with the y-axes expressed as the cumulative rate of
reaction, including all five precursors. These are reactions that produce Br atoms from stable
reservoir species, which is an important distinction from the propagation reactions that produce
Br atoms through radical reactions. $Br_2$ photolysis is calculated as 2 x $J_{Br2}$[$Br_2$]. Here, we do not
separate $Br_2$ produced in the gas-phase versus that directly emitted from a surface (this will be
discussed further in Section 3.5). The contribution of $Br_2$ photolysis in producing Br atoms vastly
dominates the cumulative production rate (Figure 6A). Therefore, in Figure 6B we show the
initiation terms without $Br_2$ photolysis so that these other production pathways can be visualized.

Effectively, $Br_2$ photolysis alone controls the production of bromine atoms, while the

remaining initiation pathways combined add only a minor contribution. Among the minor
pathways, HOBr photolysis is the most significant during non-ODE days, with the exception of
the high $NO_x$ period of March 25, where $BrNO_2$ has the largest impact. In a highly polluted
environment, halogen cycling through $NO_x$ reservoirs would become significantly more
important, as has been observed with $ClNO_2$ in mid-latitude regions (Thornton et al., 2010;
Mielke et al., 2011; Young et al., 2012). The small contribution of HOBr photolysis to bromine
atom production is an important point, because the gas-phase BrO + HOBr $\rightarrow$ BrO + $HO_2$ ozone
depletion cycle (that proceeds via HOBr photolysis rather than surface deposition) has been
considered to be significant previously (see, e.g., Hausmann and Platt, 1994), though Zeng et al.
(2006) note that HOBr photolysis has only a small effect on $BrO_x$ cycling. Using the version of
our model that is unconstrained to HOBr, but incorporates a larger surface deposition in order to
reproduce observations (Figure 2C), we were able to determine that photolysis accounts for 19%
of the HOBr sink integrated over the 7-day simulation period. Surface deposition accounts for
80%, and other known gas-phase reactions (HOBr + Br, HOBr + Cl, HOBr + OH, HOBr + O)
are only minor sink terms at a combined 1%.

The cumulative rates of reaction of the most important propagation pathways, with and

without iodine, are shown in Figure 7 A and B. The rate of the BrO + BrO reaction is calculated
as $2k[BrO]^2$, since this reaction results in the production of two Br atoms. The reaction pathways
that dominate the bromine propagation, i.e., BrO photolysis and reaction with NO, are those that
do not result in a net ozone loss. This has been previously recognized and applied to Br steady-
state calculations in several works (e.g., Platt and Janssen, 1995; Zeng et al., 2006; Holmes et al.,
2010), and demonstrates that much of the time BrO regenerates Br without a net loss of ozone
for the simulated conditions in Barrow.  Indeed, in our previous paper, we calculated that ~70%
of gas-phase BrO reforms ozone via photolysis or reaction with NO over this period (Thompson
et al., 2015). The inset pie charts, which show the average fractional importance of the various
propagation reactions for 29 and 30 March, reveal that these two pathways account for 88 – 91%
of the total.   Interestingly, the BrO self-reaction is small in comparison, with an average
contribution of 5 – 6%, and a maximum of 46%.  However, if we consider only those reactions
that *do* lead to a net ozone loss, then the BrO self-reaction accounts for an average of 71% and a
maximum of 98% of the propagation. The rate of the BrO + ClO reaction rate is much smaller
than that for BrO + BrO, though not insignificant. While on average this reaction pathway
accounts for only 2%, it does reach 16% when $Cl_2$ is high on 29 March.  In considering only

617 those reactions that result in a net ozone loss, the BrO + ClO pathway accounts for 21% on

618 average, and up to a maximum of 57%. In Panel B, the Base with Iodine scenario is shown. At

619 these levels, the BrO + IO reaction accounts for 4% of the propagation, which is at times

620 comparable to BrO + BrO and greater than BrO + ClO, even at the low IO mixing ratios in this

621 simulation (~1 pptv).

622  The short gas-phase chain length calculated for bromine propagation indicates that there

623 are large reactive bromine ($BrO_x$) sinks terminating the chain reaction. Figure 8 presents the

624 rates of the most important $BrO_x$ termination reactions, with the y-axis expressed as the

625 cumulative rate of reaction. Here it can be seen that reaction of BrO with $NO_2$ is the dominant

626 sink for $BrO_x$ on non-ODE days for the conditions encountered at Barrow, while Br reaction with

627 $CH_3CHO$ is most important when $O_3$ is depleted. That $HO_2$ is a significant sink, and would be

628 more so in less anthropogenically-impacted Polar Regions, points toward the importance of

629 heterogeneous recycling through the bromine explosion mechanism. During ozone depletion,

630 such as the major event from days $26 - 28$ March ($[O_3] < 5$ppbv) when BrO is mostly absent,

631 $CH_3CHO$ becomes the primary sink term for Br, and HCHO is relatively more important. The

632 strength of the $CH_3CHO$ sink is much greater than is HCHO, as noted previously by Shepson et

633 al. (1996) and Bottenheim et al. (1990). Of note are the relatively similar magnitudes for the

634 total rate of reaction of the initiation, propagation, and termination reactions shown in Figures 6,

635 7, and 8, respectively, which of course must be the case for a chain length near 1. This accounts

636 for the short bromine chain length determined here and also indicates that to sustain elevated

637 bromine radical concentrations necessary to deplete $O_3$ requires a relatively large $Br_2$ source,

638 likely in the form of a significant flux of $Br_2$ from the snow surface, or from in-situ production

639 from aerosols.


### 3.4 Ozone loss rate

Since the chain length calculations suggest a larger than expected contribution of
heterogeneous bromine recycling to Br atom production, to examine this further, we calculated
the rate of net ozone loss by Br and Cl in the Base Model using Equation 7 and compared this
rate to that using the estimation method presented in previous works as shown in Equation 2
(Platt and Janssen, 1995; Le Bras and Platt, 1995). Additionally, the total simulated chemical
ozone loss in the Base Model was calculated from Equation 8, which includes $O_3$ destruction by
OH, $HO_2$, and photolysis (determined here as $k[O(^1D)][H_2O]$).
$O_3\ Loss\ by\ Br\ and\ Cl\ =\ (k[Br][O_3] - J[BrO] - k[BrO][NO])$         (7)
$+\ (k[Cl][O_3] - J[ClO] - k[ClO][NO])$
$Total\ Chemical\ O_3\ loss\ rate\ =\ k[Br][O_3] + k[Cl][O_3] + k[O(^1D)][H_2O]$    (8)
$+\ k[OH][O_3] + k[HO_2][O_3] - k[BrO][NO]$
$-\ J[BrO] - k[ClO][NO] - J[ClO]$
The method in Equation 2 assumes that the rate of ozone loss is equivalent to the rate at which
Br is regenerated through BrO reaction with itself and ClO (thus assuming efficient gas-phase
propagation and a long chain length), whereas Equation 7 accounts for all net ozone destruction
by Br and Cl, by correcting for those reactions that release a triplet oxygen atom and reform $O_3$.
In other words, this method accounts for the fact that some BrO radicals react to terminate the
chain (and at steady state, an equivalent $BrO_x$ production rate is necessary). Figure 9A compares
these two estimations for $O_3$ loss rate in green (Equation 2) and pink (Equation 7). This
comparison clearly shows that there is a large difference between the methods, with the
estimation from Equation 2 significantly smaller overall. Additionally, the total chemical $O_3$ loss
(calculated by Equation 8) is shown in the dashed black trace. The $O_3$ loss rate estimation
presented in Equation 7 accounts for nearly all of the chemical $O_3$ loss (i.e., most chemical $O_3$
loss is a result of halogen chemistry), such that the dashed black line lies nearly perfectly on top
of the pink shaded regions.

In Figure 9B, we show a regression of the two estimation methods (Equation 2 in green

and Equation 7 in pink) versus the total chemical $O_3$ loss rate (Equation 8). Here it can be seen
from the pink data that halogen chemistry accounts for 99% of the total chemical $O_3$ loss under
the conditions simulated here. Importantly, the $O_3$ loss rate estimation presented in Equation 2
accounts for only 44% of the total chemical $O_3$ loss rate.

In the 1994 work by Hausmann and Platt, the authors also considered the BrO + HO $\rightarrow$

Br + $HO_2$ gas-phase ozone depletion cycle as a proxy for estimating the $O_3$ loss rate, using the
equation shown below (Equation 17 of Hausmann and Platt, 1994).
$$-\frac{d[O_3]}{dt} = (k_5 \cdot [\text{BrO}] \cdot [\text{HO}_2]) \qquad\qquad\qquad\qquad (9)$$
The authors only considered the gas-phase cycle of HOBr here with the photolysis of HOBr
regenerating Br. At the time of this publication, the heterogeneous cycling of HOBr had only
recently been proposed and had not been fully validated. Hausmann and Platt showed that
Equation 9 resulted in a significantly lower estimation for $O_3$ depletion than did Equation 1,
which considered only the BrO-BrO cycle. In Figure 9B, we show also the $O_3$ loss rate estimated
using Equation 9 in blue. Our results corroborate that of Hausmann and Platt (1994), and
demonstrate that Equation 9 can account for only 18% of the $O_3$ loss. To examine this one step
further, we present an additional regression in Figure 8B (orange trace) that combines Equations
2 and 9, thereby considering the three predominant gas-phase $O_3$ depletion cycles of BrO-BrO,
BrO-ClO, and BrO-$HO_2$. This still can only account for 60% of the $O_3$ loss.

Our analysis quantitatively expresses the conclusion that the gas-phase recycling of

bromine is not as efficient as previously considered and that it is often the case, for Barrow, that
BrO$_x$ terminations must often, through reactions R5 or R10, be followed by heterogeneous
production of Br$_2$ through condensed-phase reactions of HOBr and/or BrONO$_2$. In other words,
the reproduction of Br$_2$ via reactive deposition/uptake of HOBr and/or BrONO$_2$ onto surfaces,
followed by their gas-phase production via BrO + HO$_2$ and BrO + NO$_2$, respectively, plays a
significant role in the catalytic ozone loss involving Br atoms in the Arctic boundary layer. An
important conclusion from this analysis is that the chemical O$_3$ loss rate is largely underestimated
when calculated from only BrO observations using the previously accepted $2(k[\text{BrO}]^2 +$
[BrO][ClO]) method, and one should be cautious about drawing conclusions about O$_3$ depletion
rates and timescales based solely on BrO observations.  This may have significant impacts on the
process of examining ODEs and addressing the extent to which they represent local scale
chemistry versus transport effects. While this situation is significantly impacted by local NO$_x$
sources at Barrow, NO$_x$ is expected to increase with development around the Arctic.

**3.5  Bromine atom production**

If it is the case that heterogeneous recycling is of such importance, it may be that

Reaction R5 (BrO + HO$_2$) competes favorably with Reaction R3 (BrO + BrO).  Panel A of
Figure 10 shows the rates of reactions R5 and R3.  This plot demonstrates that for our modeling
results the rate of reaction of BrO with HO$_2$ is often of a comparable or greater magnitude than
the BrO self-reaction, and remains significant throughout the simulated period.  Previous
modeling work by Sander et al. (1997) also compared the rates of these two critical reactions
(Figure 2 of that work). In contrast to our results, their model predicted that the rate of the BrO +
BrO reaction was up to a factor of 8 greater than that of BrO + HO$_2$. The reason for this
difference may perhaps be the much lower mixing ratios of HO$_2$ in the model by Sander and
coworkers. Their model predicted $HO_2$ daily maxima of 0.2 to 0.6 pptv for most days, increasing
to 1.8 pptv on the final three days of their simulation. In contrast, $HO_2$ observations at Barrow
were frequently greater than 5 pptv and up to 10 pptv. As demonstrated in Thompson et al.
(2015), HCHO was a dominant factor in controlling the $HO_2$ mixing ratios in Barrow. The low
levels of $HO_2$ in Sander et al. (1997) likely also contribute to their low predicted HOBr mixing
ratios, which do not exceed 1 pptv in their model. This also is much lower than observations at
Barrow, where HOBr reaching 10 pptv to 20 pptv was measured during our simulated period.
Because the $BrO + HO_2$ reaction is of primary importance for the bromine explosion mechanism,
our result supports the hypothesis that heterogeneous recycling may be equally or even more
important than gas-phase recycling of reactive bromine.

Given that the chain length is small, it must be that initiation is an important source of Br

atoms in order to sustain BrO and lead to $O_3$ depletion. To further examine the question of
surface emissions/release versus gas-phase recycling, we determined the rate of production of Br
atoms via photolysis of $Br_2$ and BrCl (Equation 10) compared to the rate of production of Br
atoms through gas-phase recycling calculated by Equation 11.  Because our model is constrained
by $Br_2$ observations and we do not produce $Br_2$ from surfaces via heterogeneous reactions, the
photolysis of $Br_2$ includes $Br_2$ that is both emitted from surfaces and that is formed via gas-phase
reactions. To correct for the $Br_2$ that is formed in the gas-phase reactions so that Equation 10
represents our best approximation for surface-emitted $Br_2$, we created a proxy in the model, $Br_2$*,
that represents the $Br_2$ produced from gas phase reactions. These reactions include $Br + BrNO_2$,
$Br + BrONO_2$, and the $BrO + BrO$ branch that produces $Br_2$. Equation 10 is thus corrected for the
gas-phase generated $Br_2$ by subtracting the photolysis of $Br_2$*. A comparison of $Br_2$ and $Br_2$*
reveals that these three gas-phase production pathways account for an average of 35% of
observed $Br_2$, suggesting that the snowpack and/or aerosols emits the remaining 65%. Again, we
cannot distinguish between snow or aerosol production using this method.

*Br Production from Surface-derived Br$_2$, BrCl* = 2 x $J_{Br2}$[Br$_2$] + $J_{BrCl}$[BrCl] - 2 x $J_{Br2}$[Br$_2$*]    (10)
*Br Production via Gas-phase Recycling* =  2$k$[BrO][BrO] + $k$[BrO][ClO]                          (11)
+ $k$[BrO][NO] + $k$[BrO][OH]+ $k$[BrO][O($^3$P)]
+ $k$[BrO][CH$_3$OO] + $k$[BrO][CH$_3$COOO]
+ $J_{HOBr}$[HOBr] + $J_{BrO}$[BrO] + $J_{BrONO2}$[BrONO$_2$]
+ $J_{BrNO2}$[BrNO$_2$]


Panel B of Figure 10 compares the results of Equations 10 and 11, showing the total rate of Br
atom production separated into Br production from the derived "surface-emitted" Br$_2$ and BrCl
(purple) and from gas-phase Br recycling (orange); Panel C plots the fraction of total Br atom
production that is due to production from Br$_2$ and BrCl surface emissions/release.  The majority
of the time during this 7-day period Br atom production from Br$_2$ and BrCl emissions/release
(Equation 10) accounts for 30% or greater of the total, and at times reaches up to 90%.  This
explains both how ozone depletion can be rapid despite the short calculated bromine radical
chain length, as well as the difference found between the two methods of estimating O$_3$ loss rate
in Figure 9.  It is concluded from this analysis, then, that the condensed phase recycling of
bromine can be of equal or greater importance to the evolution of ODEs than gas-phase Br
regeneration through radical recycling reactions.

**4 Conclusions**

The analysis presented here suggests that the gas-phase recycling of bromine species may

be less important than commonly believed, and we conclude that heterogeneous recycling is
critical for the evolution of ODEs/AMDEs, consistent with results by Michalowski et al. (2000),
Piot and von Glasow (2008), and Toyota et al. (2011, 2014). Expressed in another way, the
reproduction of $Br_2$ via reactive deposition/uptake of HOBr and/or $BrONO_2$ onto surfaces,
followed by their gas-phase production via $BrO + HO_2$ and $BrO + NO_2$, respectively, is critical
for sustaining the Br atom chemistry leading to $O_3$ depletion in the Arctic boundary layer.

To support this conclusion, we have used the gas-phase bromine chain length, which has

not previously been applied to Arctic halogen chemistry, as an objective metric. The gas-phase
bromine chain length is much shorter than expected, suggesting that much of the Br present in
the gas-phase is Br from surface emissions/release.  Again note that our calculation of chain
length includes photolysis of BrO and $BrO + NO$, which do not result in net $O_3$ loss. Had we
omitted these two reactions, which we have found are in fact dominating the radical propagation,
the chain length would be, on average, 80% shorter. Because of the small chain length calculated
for Br, one must be cautious about drawing conclusions about $O_3$ depletion from BrO
measurements alone. We recommend concurrent measurements of a broad suite of inorganic
bromine species for accurate study of these ozone depletion events. The very low mixing ratios
of HOBr predicted by Sander et al. (1997) and the high mixing ratios originally predicted by our
model point to the need for measurements of these species to validate the accuracy of Arctic
models.

We find that between 30 – 90% of Br atoms are produced from surface emissions/release

of $Br_2$ and BrCl, though we cannot distinguish snow sources from aerosol sources using our
model.  However, it is important to note that we do not know how much of the condensed phase
$Br_2$ production derives from reaction R7, or from some other condensed phase process, e.g.
oxidation of $Br^-$ by OH radicals (Abbatt et al., 2010). The in situ snow chamber experiments by
Pratt et al. (2013) demonstrate a strong $Br_2$ source from the snowpack; similar field observations
proving significant $Br_2$ release from Arctic aerosol are currently lacking.  If the snow surface is
the primary sources of these emissions, then a strong vertical gradient would be expected in the
near surface boundary layer, and our estimations for the Br chain length would be only valid for
the height of our measurements ($\sim 1$ m above the snow). Strong deposition to the snow would
also induce a vertical gradient in these species. If, however, aerosols are an important source of
$Br_2$ (or other halogen precursors), then $Br_2$ production should occur throughout the entire height
of the boundary with no significant vertical gradient, in a similar fashion as has been observed
for $ClNO_2$, which is a known product of aerosol chemistry (Young et al., 2012).  It is clear that
vertically-resolved measurements of these halogen precursors are imperative for our
understanding of halogen production in the Arctic.

The production of $Br_2$ is quite complex and is dependent on many factors, including the

relative concentrations of bromide and chloride (among others), the availability of atmospheric
oxidants, such as ozone (e.g., Oum et al., 1998; Pratt et al., 2013), the pH of the snow surfaces or
aerosol (Toyota et al., 2011, 2014), the presence of snow phase oxidants such as $H_2O_2$ (Pratt et
al., 2013), and the replenishment of the snowpack halides from deposited sea salts.  The last of
these is governed by meteorology, the proximity of open water or saline sea ice surfaces, and
wind/storm events, making the accurate modeling of these processes very complex (Domine et
al., 2013).  Likewise, to date, it has not been possible to determine the halide concentrations or
pH of the snow grain surfaces, and these values are likely highly variable and dependent on snow
and aerosol aging and deposition of atmospheric constituents. Due to the apparent importance of
surface chemistry for both the initiation and evolution of Arctic ozone depletion events, it is clear
that more laboratory and field studies are required to decipher these complex chemical and
physical processes.   In particular, we strongly recommend studies relating to direct
measurements of surface fluxes of molecular halogens, as a function of conditions of temperature,
snowpack composition, and pH, as well as deposition velocities for the hypohalous acids (HOBr,
HOCl) to the snow. Our model overestimation of HOBr, that necessitated constraint to
observations, suggests a sometimes much stronger, but also variable, deposition of HOBr that is
currently unknown. Further, there is currently little understanding of the mechanism for $Cl_2$
production in the Arctic, and no successful measurements of IO in the High Arctic. Recent
observations of $I_2$ within the Barrow snowpack (Raso et al., 2016) suggest reactive iodine
chemistry is present in this region, and this would have an impact on Br recycling and ozone
depletion rate. Investigations into these areas would greatly increase our understanding of
halogen chemistry and ozone depletion in the Arctic.

**Acknowledgements** This work was funded by the National Science Foundation grant ARC-
0732556. Partial support for CT during preparation of this manuscript was provided by the NSF
Atmospheric and Geospace Sciences Postdoctoral Research Fellowship program. The authors
wish to thank the organizers of the OASIS 2009 field campaign, the Barrow Arctic Science
Consortium for logistics support, and all of the researchers who contributed to the campaign.
This paper is submitted in memory of our colleague and friend, Roland von Glasow.

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

**Table 1.** Reactions used in the model that are pertinent to bromine chemistry.  All rate constants
(with the exception of photolysis $J$ coefficients) are in units of $cm^3$ molecule$^{-1}$ s$^{-1}$.

| Gas-Phase Reactions | Rate Constant | Reference |
|---|---|---|
| Br + $O_3$ → BrO | 6.75 x 10$^{-13}$ | *Atkinson et al.* [2004] |
| Br + $C_2H_4$ → HBr + $C_2H_5OO$ | 1.3 x 10$^{-13}$ | *Atkinson et al.* [2004] |
| Br + $C_3H_6$ → HBr + $C_3H_5$ | 1.60 x 10$^{-12}$ | *Atkinson et al.* [2004] |
| Br + HCHO → HBr + CO + $HO_2$ | 6.75 x 10$^{-13}$ | *Sander et al.* [2006] |
| Br + $CH_3CHO$ → HBr + $CH_3COOO$ | 2.8 x 10$^{-12}$ | *Atkinson et al.* [2004] |
| Br + $C_3H_6O$ → HBr | 9.7 x 10$^{-12}$ | *Wallington et al.* [1989] |
| Br + nButanal → HBr | 9.7 x 10$^{-12}$ | estimate from *Michalowski et al.* [2000] |
| Br + $CH_3OOH$ → HBr + $CH_3OO$ | 4.03 x 10$^{-15}$ | *Mallard et al.* [1993] |
| Br + $NO_2$ → $BrNO_2$ | 2.7 x 10$^{-11}$ | *Atkinson et al.* [2004] |
| Br + $BrNO_3$ → $Br_2$ + $NO_3$ | 4.9 x 10$^{-11}$ | *Orlando and Tyndall* [1996] |
| Br + OClO → BrO + ClO | 1.43 x 10$^{-13}$ | *Atkinson et al.* [2004] |
| BrO + O($^3P$) → Br | 4.8 x 10$^{-11}$ | *Atkinson et al.* [2004] |
| BrO + OH → Br + $HO_2$ | 4.93 x 10$^{-11}$ | *Atkinson et al.* [2004] |
| BrO + $HO_2$ → HOBr | 3.38 x 10$^{-11}$ | *Atkinson et al.* [2004] |
| BrO + $CH_3OO$ → HOBr + $CH_2OO$ | 4.1 x 10$^{-12}$ | *Aranda et al.* [1997] |
| BrO + $CH_3OO$ → Br + HCHO + $HO_2$ | 1.6 x 10$^{-12}$ | *Aranda et al.* [1997] |
| BrO + $CH_3COOO$ → Br + $CH_3COO$ | 1.7 x 10$^{-12}$ | estimate from *Michalowski et al.* [2000] |
| BrO + $C_3H_6O$ → HOBr | 1.5 x 10$^{-14}$ | estimate from *Michalowski et al.* [2000] |
| BrO + NO → Br + $NO_2$ | 2.48 x 10$^{-11}$ | *Atkinson et al.* [2004] |
| BrO + $NO_2$ → $BrNO_3$ | 1.53 x 10$^{-11}$ | *Atkinson et al.* [2004] |
| BrO + BrO → Br + Br | 2.82 x 10$^{-12}$ | *Sander et al.* [2006] |

| 1156 | $BrO + BrO \rightarrow Br_2$ | $9.3 \times 10^{-13}$ | *Sander et al.* [2006] |
|---|---|---|---|
| 1157 | $BrO + HBr \rightarrow HOBr + Br$ | $2.1 \times 10^{-14}$ | *Hansen et al.* [1999] |
| 1158 | $HBr + OH \rightarrow Br + H_2O$ | $1.26 \times 10^{-11}$ | *Sander et al.* [2006] |
| 1159 | $CH_3Br + OH \rightarrow H_2O + Br$ | $1.27 \times 10^{-14}$ | *Atkinson et al.* [2004] |
| 1160 | $CHBr_3 + OH \rightarrow H_2O + Br$ | $1.2 \times 10^{-13}$ | *Atkinson et al.* [2004] |
| 1161 | $Cl + BrCl \leftrightarrow Br + Cl_2$ | f: $1.5 \times 10^{-11}$  r: $1.1 \times 10^{-15}$ | *Clyne et al.* [1972] |
| 1162 | $Cl + Br_2 \leftrightarrow BrCl + Br$ | f: $1.2 \times 10^{-10}$  r: $3.3 \times 10^{-15}$ | *Clyne et al.* [1972] |
| 1163 | $BrO + ClO \rightarrow Br + Cl$ | $7.04 \times 10^{-12}$ | *Atkinson et al.* [2004] |
| 1164 | $BrO + ClO \rightarrow BrCl$ | $1.15 \times 10^{-12}$ | *Atkinson et al.* [2004] |
| 1165 | $BrO + ClO \rightarrow Br + OClO$ | $9.06 \times 10^{-12}$ | *Atkinson et al.* [2004] |
| 1166 | $HOBr + OH \rightarrow BrO + H_2O$ | $5.0 \times 10^{-13}$ | *Kukui et al.* [1996] |
| 1167 | $HOBr + Cl \rightarrow BrCl + OH$ | $8.0 \times 10^{-11}$ | *Kukui et al.* [1996] |
| 1168 | $HOBr + O(^3P) \rightarrow BrO + OH$ | $2.12 \times 10^{-11}$ | *Atkinson et al.* [2004] |
| 1169 | $IO + BrO \rightarrow Br + OIO$ | $9.36 \times 10^{-11}$ | *Atkinson et al.* [2004] |
| 1170 | $IO + BrO \rightarrow IBr$ | $4.32 \times 10^{-11}$ | *Atkinson et al.* [2004] |
| 1171 | $IO + BrO \rightarrow Br + I$ | $7.2 \times 10^{-12}$ | *Atkinson et al.* [2004] |
| 1172 | | | |

| 1173 | **Photolysis Reactions** | $J_{max}$ (25 March) s$^{-1}$ | **Lifetime** | **Reference** |
|---|---|---|---|---|
| 1174 | $BrNO_3 \rightarrow Br + NO_3$ | $2.1 \times 10^{-4}$ | 1.3 h | calculated from OASIS data |
| 1175 | $BrNO_3 \rightarrow BrO + NO_2$ | $1.2 \times 10^{-3}$ | 14.2 min | calculated from OASIS data |
| 1176 | $BrO \rightarrow Br + O(^3P)$ | $3.0 \times 10^{-2}$ | 33 s | calculated from OASIS data |
| 1177 | $Br_2 \rightarrow Br + Br$ | $4.4 \times 10^{-2}$ | 23 s | calculated from OASIS data |
| 1178 | $HOBr \rightarrow Br + OH$ | $2.3 \times 10^{-3}$ | 7.2 min | calculated from OASIS data |
| 1179 | $BrNO_2 \rightarrow Br + NO_2$ | $1.5 \times 10^{-4}$ | 1.8 h | estimate from *Lehrer et al.* [2004] |
| 1180 | $BrCl \rightarrow Br + Cl$ | $1.26 \times 10^{-2}$ | 1.3 min | calculated from OASIS data |
| 1181 | | | | |
| 1182 | | | | |

| 1183 | **Mass Transfer Reactions** | $k_t$ (forward) | $k_t$ (reverse) |
|---|---|---|---|
| 1184 | $HBr_{(g)} \rightarrow H^+_{(p)} + Br^-_{(p)}$ | $1.80 \times 10^{-3}$ | |
| 1185 | $HCl_{(g)} \rightarrow H^+_{(p)} + Cl^-_{(p)}$ | $2.58 \times 10^{-3}$ | |
| 1186 | $HOBr_{(g)} \rightarrow HOBr_{(p)}$ | $1.26 \times 10^{-3}$ | |
| 1187 | $BrNO_{2(g)} \rightarrow BrNO_{2(p)}$ | $1.26 \times 10^{-3}$ | |
| 1188 | $BrONO_{3(g)} \rightarrow BrONO_{3(p)}$ | $1.26 \times 10^{-3}$ | |
| 1189 | $Br_{2(g)} \leftrightarrow Br_{2(p)}$ | $1.78 \times 10^{-5}$ | $2.97 \times 10^{8}$ |
| 1190 | $BrCl_{(g)} \leftrightarrow BrCl_{(p)}$ | $6.60 \times 10^{-4}$ | $1.91 \times 10^{10}$ |
| 1191 | $IBr_{(p)} \rightarrow IBr_{(g)}$ | $5.53 \times 10^{9}$ | |
| 1192 | $HBr_{(g)} \rightarrow H^+_{(s)} + Br^-_{(s)}$ | $1.67 \times 10^{-5}$ | |
| 1193 | $HCl_{(g)} \rightarrow H^+_{(s)} + Cl^-_{(s)}$ | $1.67 \times 10^{-5}$ | |
| 1194 | $HOBr_{(g)} \rightarrow HOBr_{(s)}$ | $1.67 \times 10^{-5}$ | |
| 1195 | $BrNO_{2(g)} \rightarrow BrNO_{2(s)}$ | $1.67 \times 10^{-4}$ | |
| 1196 | $BrONO_{3(g)} \rightarrow BrONO_{3(s)}$ | $1.26 \times 10^{-4}$ | |
| 1197 | $Br_{2(g)} \leftrightarrow Br_{2(s)}$ | $1.0 \times 10^{-5}$ | $7.71 \times 10^{-2}$ |
| 1198 | $BrCl_{(g)} \leftrightarrow BrCl_{(s)}$ | $1.25 \times 10^{-5}$ | $7.71 \times 10^{-2}$ |
| 1199 | $IBr_{(s)} \rightarrow IBr_{(g)}$ | $7.71 \times 10^{-2}$ | |
| 1200 | | | |

| 1201 | **Aqueous Phase Reactions** | $k$ (particle) | $k$ (snow) | **Reference** |
|---|---|---|---|---|
| 1202 | $Cl^- + HOBr + H^+ \rightarrow BrCl$ | $5.17 \times 10^{-21}$ | $9.30 \times 10^{-26}$ | *Wang et al.* [1994] |
| 1203 | $Br^- + HOCl + H^+ \rightarrow BrCl$ | $1.2 \times 10^{-24}$ | $2.15 \times 10^{-29}$ | *Sander et al.* [1997] |
| 1204 | $Br^- + HOBr + H^+ \rightarrow Br_2$ | $1.47 \times 10^{-20}$ | $2.64 \times 10^{-25}$ | *Beckwith et al.* [1996] |
| 1205 | $Br^- + HOI + H^+ \rightarrow IBr$ | $3.04 \times 10^{-18}$ | $5.46 \times 10^{-23}$ | *Troy et al.* [1991] |
| 1206 | $BrCl + Cl^- \rightarrow BrCl_2^-$ | 3.3 | $5.99 \times 10^{-5}$ | *Michalowski et al.* [2000] |
| 1207 | $BrCl_2^- \rightarrow BrCl + Cl^-$ | $1.58 \times 10^{9}$ | $1.58 \times 10^{9}$ | *Michalowski et al.* [2000] |
| 1208 | $BrCl + Br^- \rightarrow Br_2Cl^-$ | 3.3 | $5.99 \times 10^{-5}$ | *Michalowski et al.* [2000] |
| 1209 | $Br_2Cl^- \rightarrow BrCl + Br^-$ | $3.34 \times 10^{5}$ | $3.34 \times 10^{5}$ | *Wang et al.* [1994] |
| 1210 | $Cl_2 + Br^- \rightarrow BrCl_2^-$ | 4.27 | $7.66 \times 10^{-5}$ | *Wang et al.* [1994] |
| 1211 | $BrCl_2^- \rightarrow Cl_2 + Br^-$ | $6.94 \times 10^{2}$ | $6.94 \times 10^{2}$ | *Wang et al.* [1994] |

$O_3 + Br^- \rightarrow HOBr$                      4.5 x $10^{-9}$              8.08 x $10^{-14}$        *Oum et al.* [1998]

**Table 2.**  Median mid-day bromine chain lengths for 25, 29, and 30 March 2009 (days with $O_3$
present) determined for four different modeling scenarios with different combinations of
halogens present.  Method 1 refers to Equation 3 (using terminations reactions) and Method 2
refers to Equation 4 (using initiation reactions).

| | 25 March | | 29 March | | 30 March | | Average (1-σ st. deviation) | |
|---|---|---|---|---|---|---|---|---|
| | Method 1 | Method 2 | Method 1 | Method 2 | Method 1 | Method 2 | Method 1 | Method 2 |
| Br only | 1.25 | 0.85 | 1.51 | 1.10 | 1.79 | 1.40 | 1.52 (± 0.27) | 1.11 (± 0.28) |
| Br and Cl (Base) | 1.29 | 0.84 | 1.43 | 1.03 | 1.58 | 1.29 | 1.43 (± 0.14) | 1.05 (± 0.22) |
| Br and Low I | 1.37 | 0.86 | 1.60 | 1.12 | 1.82 | 1.41 | 1.59 (± 0.22) | 1.13 (± 0.28) |
| Br, Cl, and I | 1.37 | 0.87 | 1.51 | 1.04 | 1.65 | 1.31 | 1.51 (± 0.14) | 1.07 (± 0.23) |


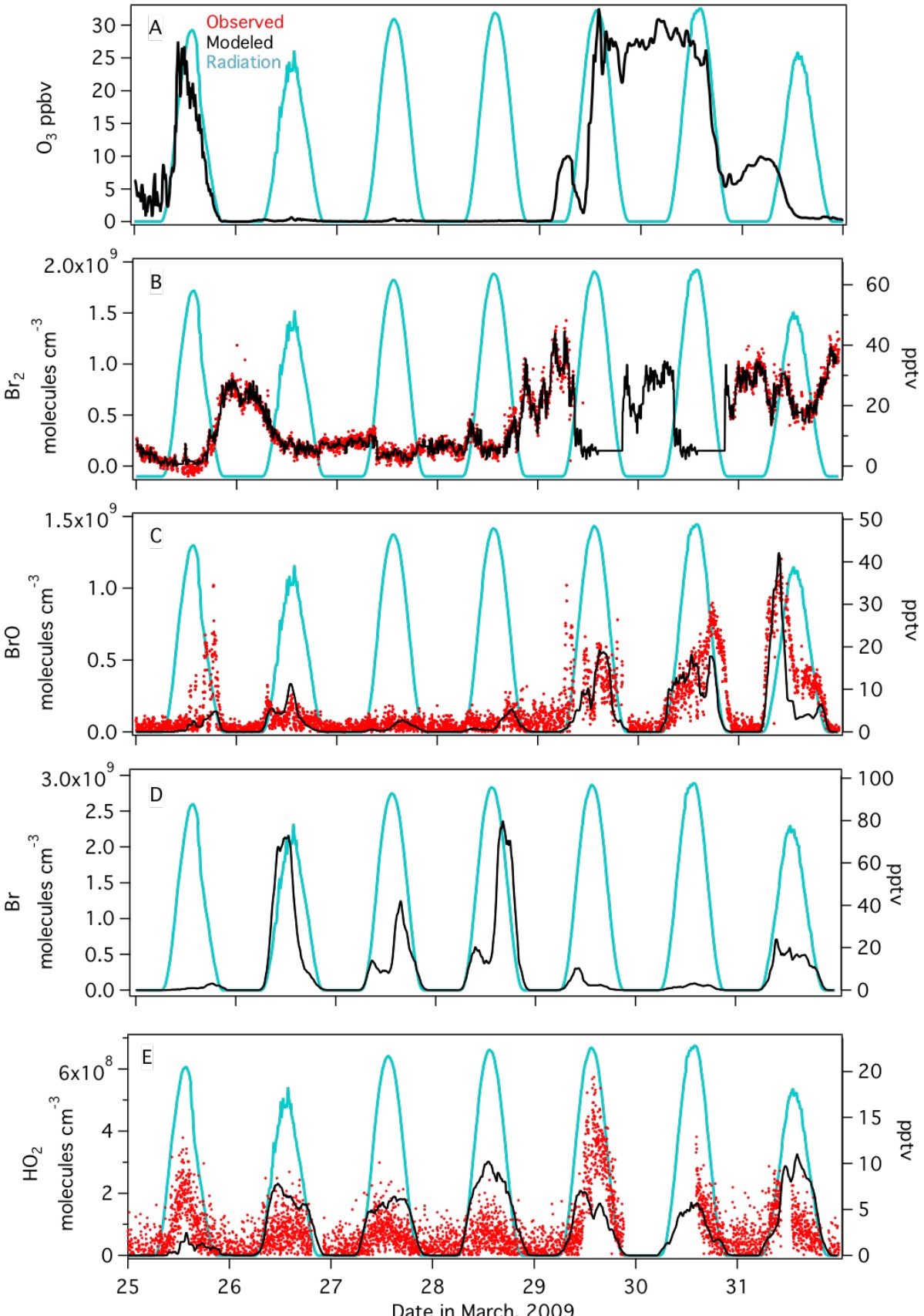

**Fig 1.** Time-series of gas-phase concentrations and mixing ratios of $O_3$, $Br_2$, BrO, Br, and $HO_2$ in
the model (black trace) for the seven-day period simulated. Observations are plotted in red
where available for $Br_2$, BrO, and $HO_2$. $O_3$ and $Br_2$ are constrained species in the model.
Simulated output of BrO, Br, and $HO_2$ are smoothed by hourly averaging. Radiation is shown as
the cyan trace as a reference. Time is expressed in Alaska Standard Time.

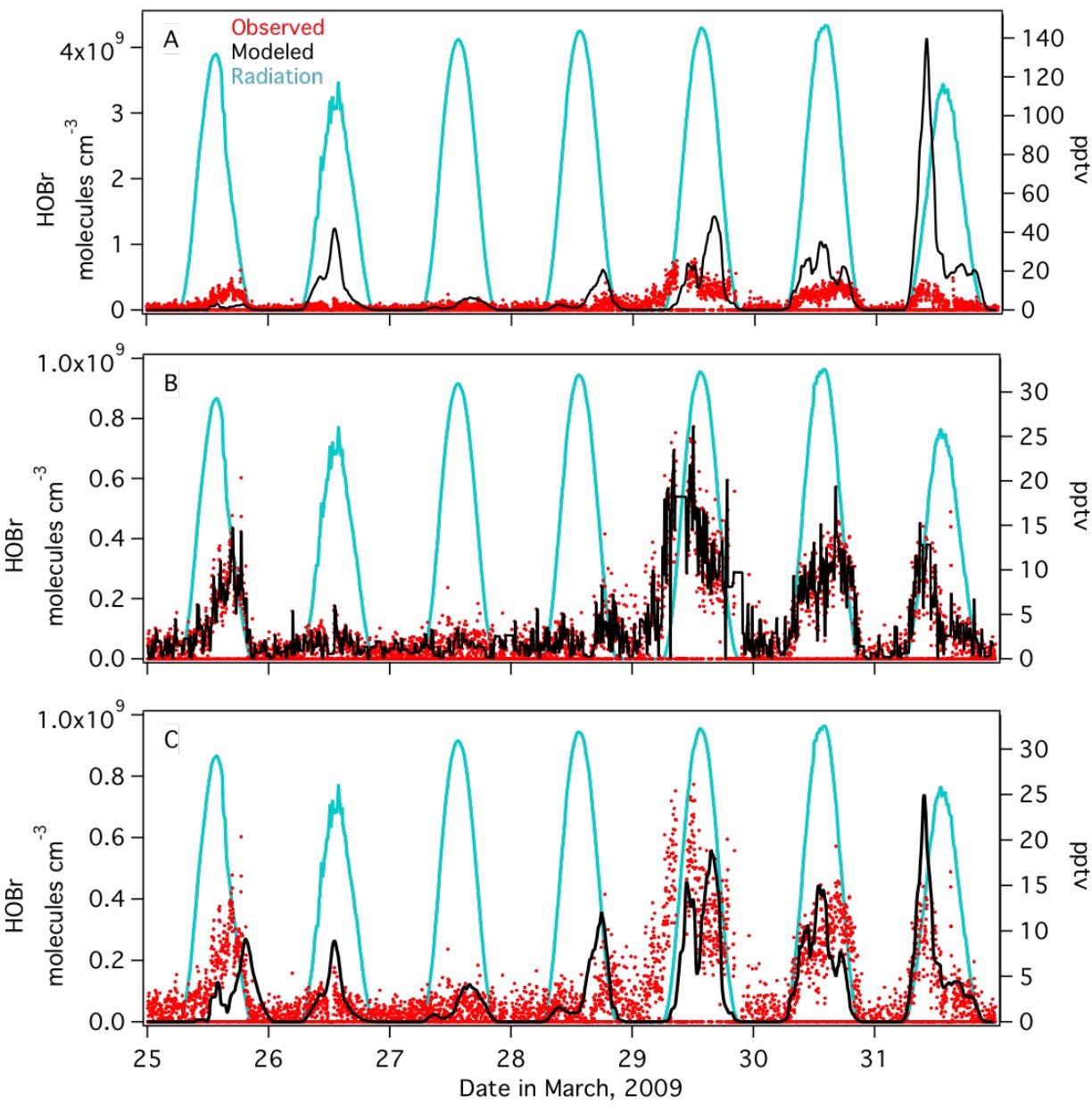

**Fig 2.** Simulated (black trace) versus observed (red markers) HOBr mixing ratios shown for
three different versions of the model: A) HOBr unconstrained and allowed to freely evolve with
a constant surface deposition term as described in the Methods, B) HOBr constrained to
observations, C) HOBr unconstrained but with a variable surface deposition that is enhanced
during higher wind speeds. Simulated (unconstrained) output in Panels A and C are smoothed by
hourly averaging. Radiation is shown as the cyan trace as a reference.  Time is expressed in
Alaska Standard Time.

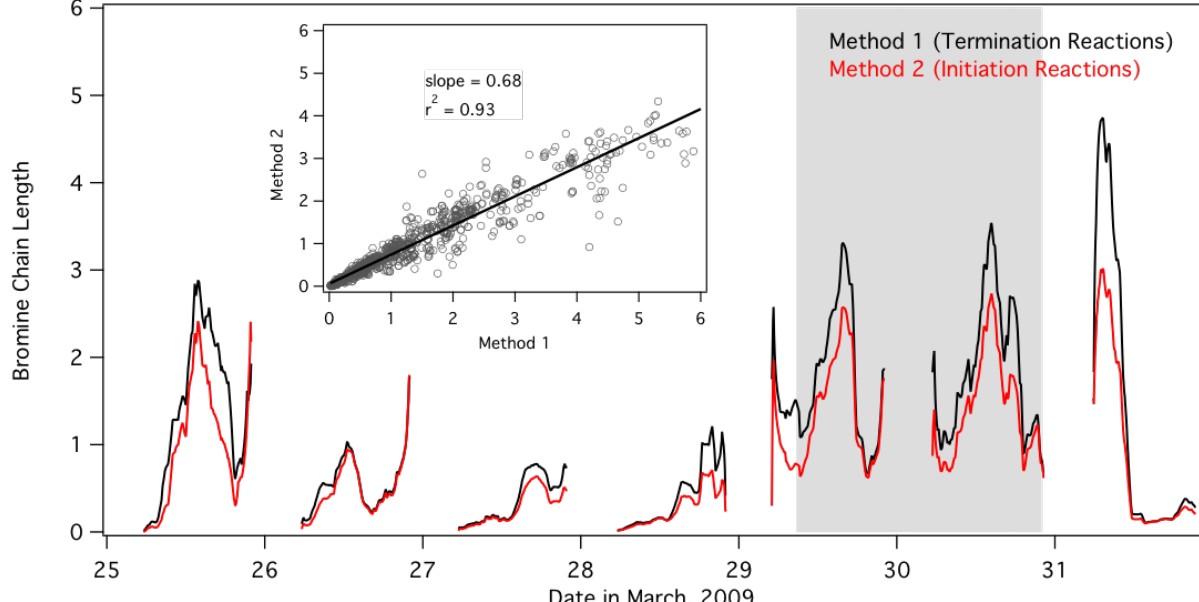


**Fig 3.**  Time-series of model calculated bromine chain length for the daytime hours (7:00 to
21:00 AKST).  Method 1 is plotted as the black trace and Method 2 is plotted as the red trace.
Model output is smoothed by hourly averaging. The grey shaded box represents a period of
missing $Br_2$ observations. The inset graph shows a linear regression of Method 1 and Method 2
calculations. Time is expressed in Alaska Standard Time.

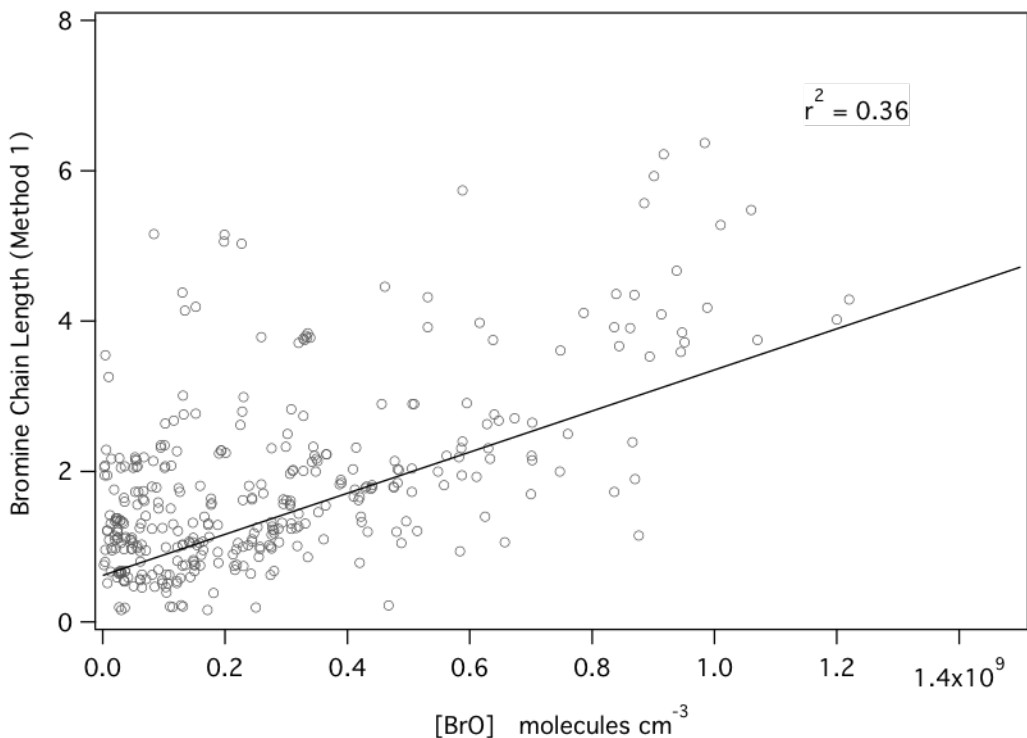


**Fig 4.** Regression of daytime (7:00 – 21:00 AKST) bromine chain length calculated by Method
1 (Equation 5) and simulated BrO concentration.

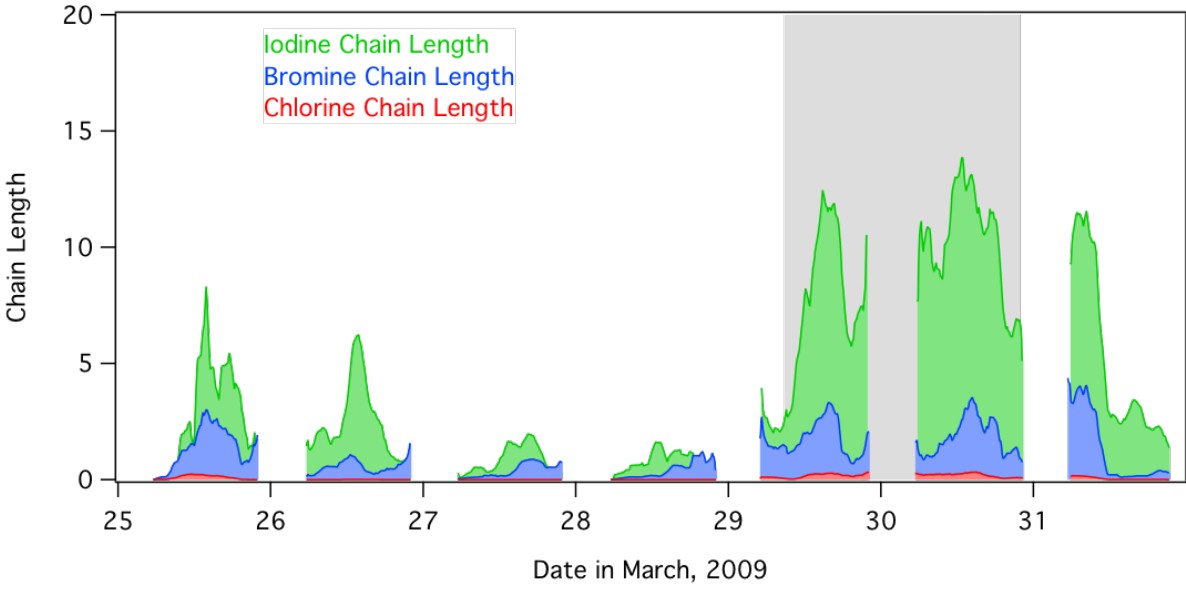


**Fig 5.** Calculated chain lengths for iodine (green), bromine (blue), and chlorine (red) across the
seven days of the simulated period modeled using the Base + Iodine scenario. Model output is
smoothed by hourly averaging. The grey shaded box represents a period of missing $Br_2$
observations. Time is expressed in Alaska Standard Time.

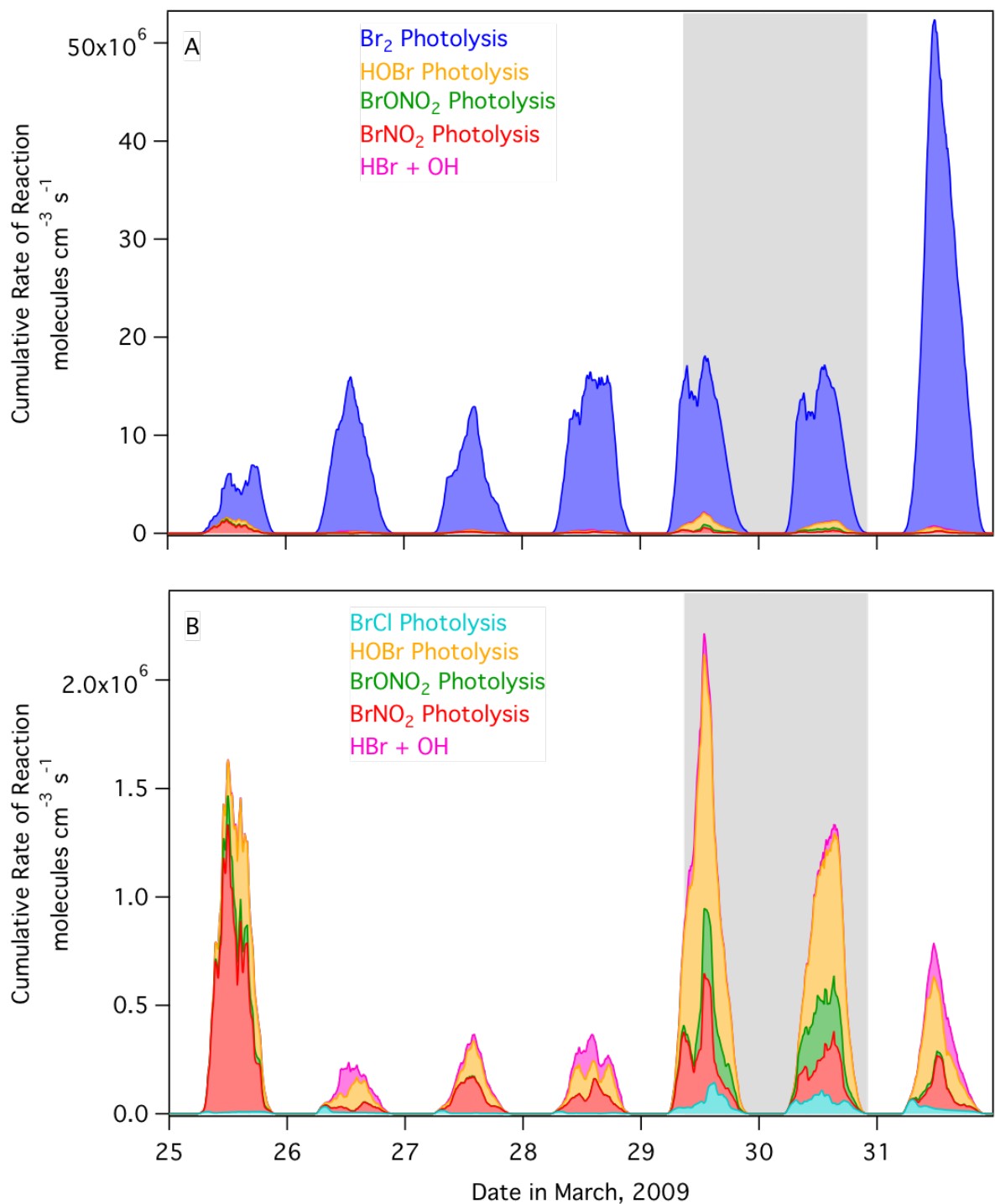


**Fig 6.** Time-varying rates of the most important bromine initiation reactions in the Base Model.
Panel A includes photolysis of $Br_2$, which dominates the bromine initiation. $Br_2$ photolysis is
calculated as 2 x $J_{Br2}[Br_2]$. In Panel B, $Br_2$ photolysis has been removed so that the minor terms
can be visualized. Panel B also includes BrCl, which contributes only a negligible amount to
bromine initiation. Model output is smoothed by hourly averaging. The y-axis is expressed as a
cumulative rate of reaction. The grey shaded box represents a period of missing $Br_2$ observations.
Time is expressed in Alaska Standard Time.

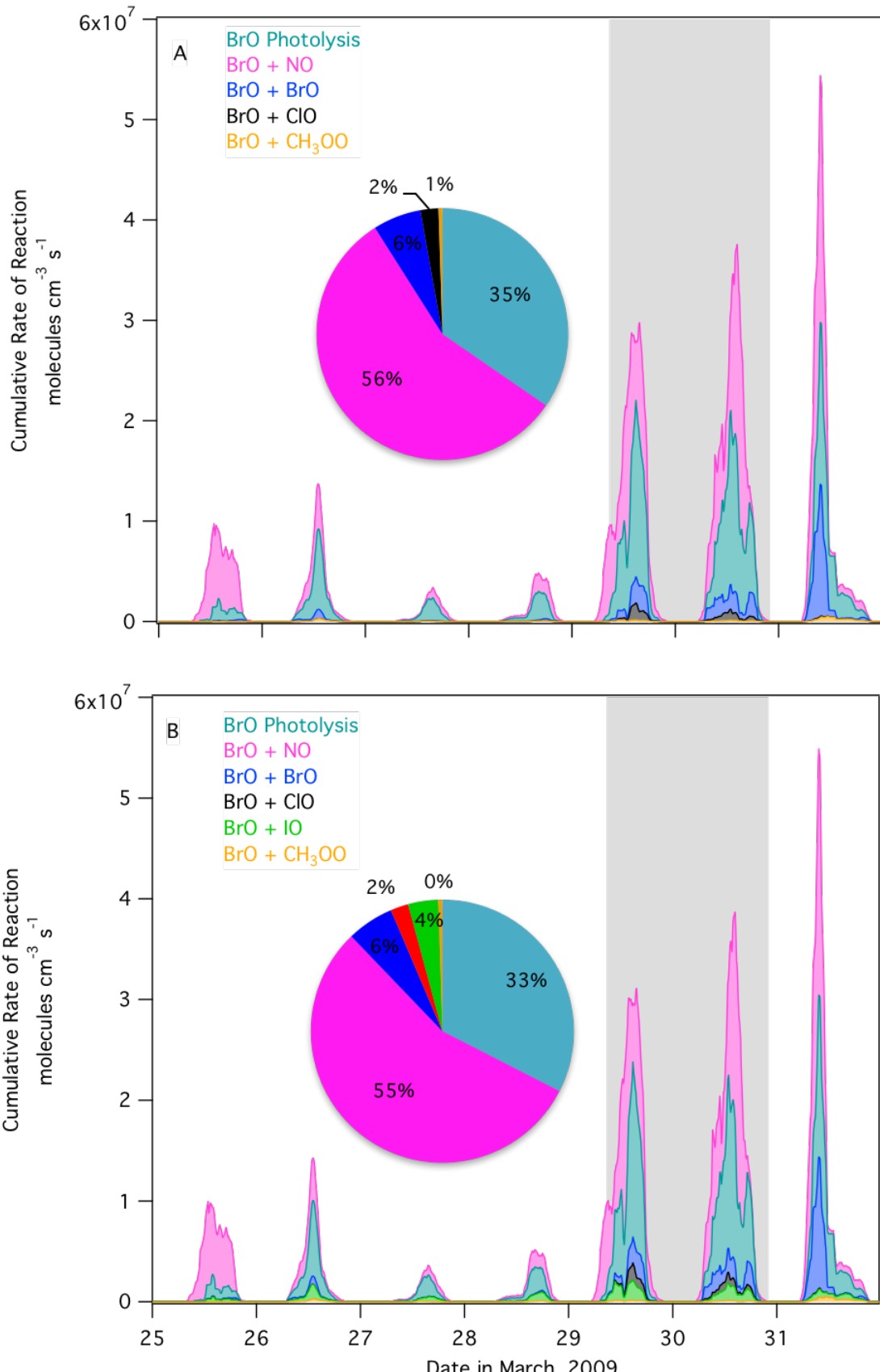

**Fig 7.**   Time-varying rates of the most important bromine propagation reactions in the Base
Model with Br and Cl present (Panel A) and with iodine included (Panel B). The BrO + BrO
reaction is calculated as $2k[\mathrm{BrO}]^2$ as this reaction regenerates two Br atoms. Model output is
smoothed by hourly averaging. The y-axis is expressed as a cumulative rate of reaction. The grey
shaded box represents a period of missing $\mathrm{Br_2}$ observations. Time is expressed in Alaska
Standard Time. The inset pie charts show the average fractional importance of each reaction
pathway for only days 29 and 30 March (i.e. background $\mathrm{O_3}$ days).

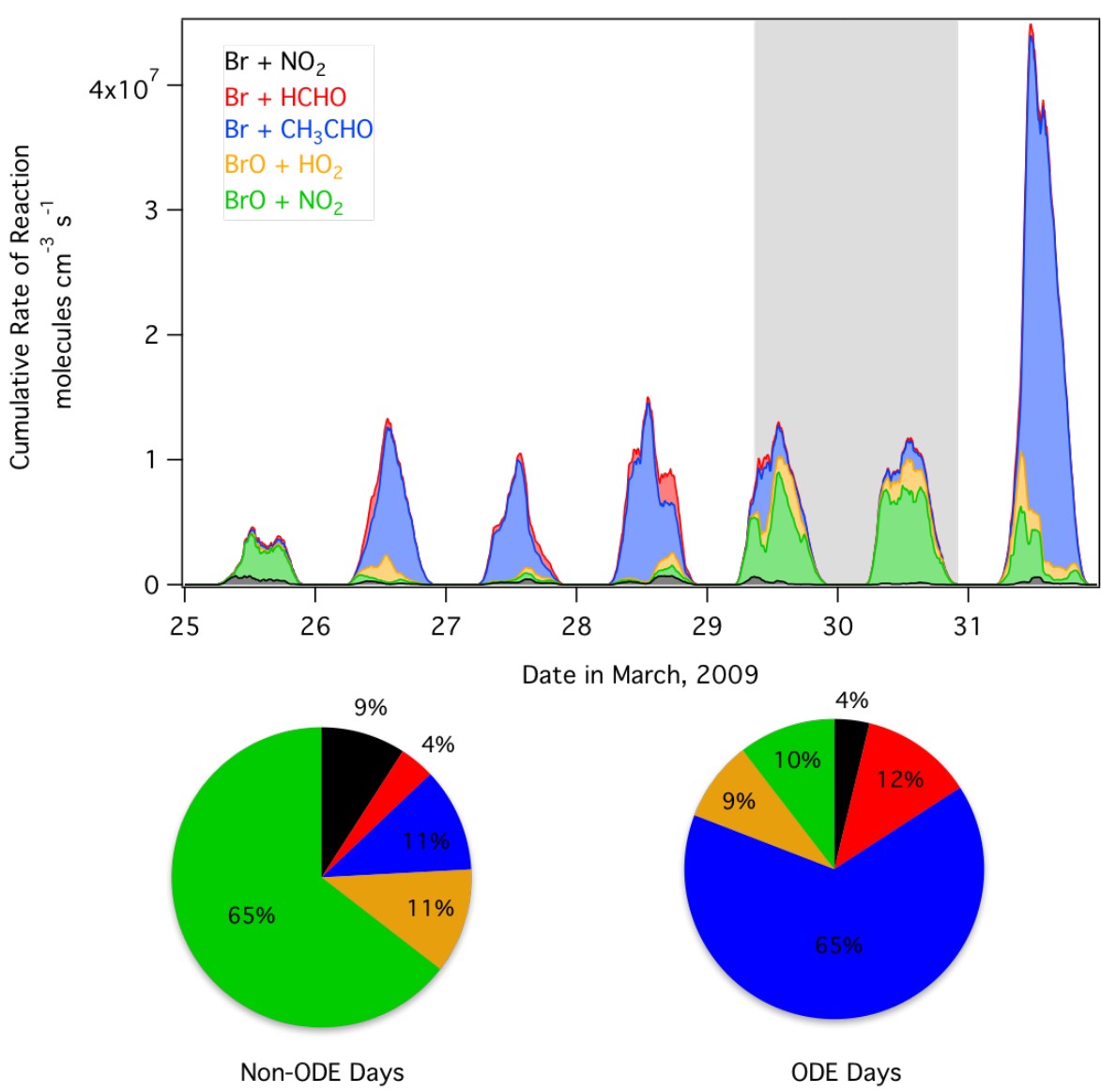


**Fig 8.**   Time-varying rates of the most important reactive bromine ($\mathrm{BrO_x}$) termination reactions
in the Base Model.  Model output is smoothed by hourly averaging. The y-axis is expressed as a
cumulative rate of reaction.  The grey shaded box represents a period of missing $\mathrm{Br_2}$ observations.
Time is expressed in Alaska Standard Time. The pie charts show the average fractional
importance of each reactive bromine sink for non-ODE (background $\mathrm{O_3}$) days and ODE days.

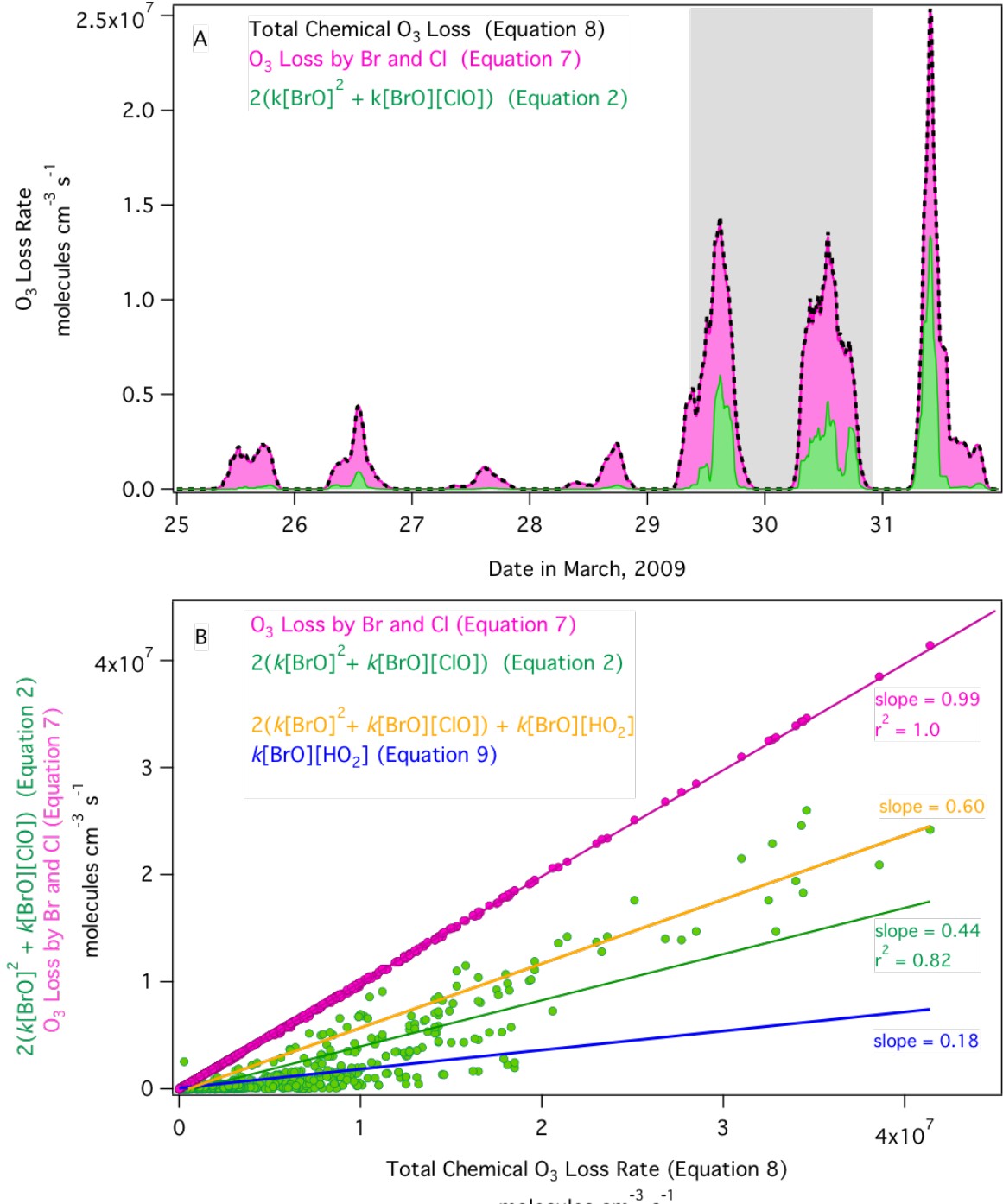


**Fig 9.** A) Comparison of the time-varying $O_3$ loss rate calculated using the estimation of
$2(k[BrO]^2 + k[BrO][ClO])$ (Equation 2, green), the simulated $O_3$ loss rate by Br and Cl (Equation
7, pink), and the total simulated chemical $O_3$ loss rate (Equation 8, dashed black trace). Model
output is smoothed by hourly averaging. The grey shaded box represents a period of missing $Br_2$
observations. Time is expressed in Alaska Standard Time. B) Shown is a regression of the
$2(k[BrO]^2 + [BrO][ClO])$ estimation method (Equation 2) versus the total simulated chemical $O_3$
loss rate in the Base Model (Equation 8) in the green data, and a regression of $O_3$ loss rate by Br
and Cl only (Equation 7) versus the total simulated chemical $O_3$ loss rate in the pink data. The
blue trace represents the $O_3$ loss rate estimated by only considering $k[BrO][HO_2]$ (Equation 9).
The orange trace estimates $O_3$ loss rate combining the three major gas-phase ozone depletion
cycles. The slopes represent the fraction of the chemical $O_3$ loss rate that can be accounted for by
each method.  For the conditions simulated, the commonly used estimation method of $2(k[BrO]^2$
$+ [BrO][ClO])$ only accounts for 44% of the chemical $O_3$ loss rate.

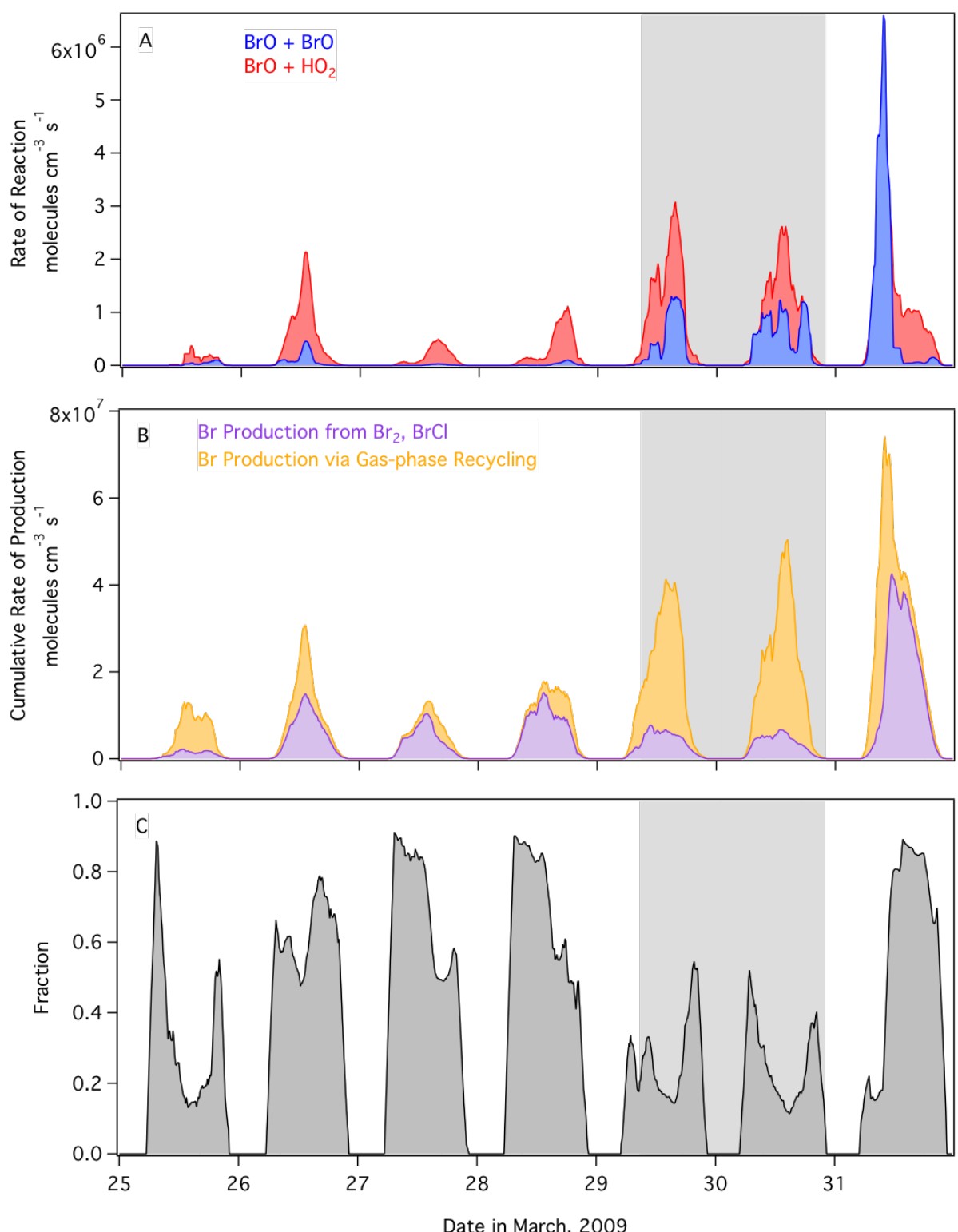


**Fig 10.** Panel A: Comparison of the rate of reaction of BrO + BrO (blue) and BrO + HO$_2$ (red).
Panel B: The cumulative rate of Br atom production separated into the Br production rate from
the photolysis of Br$_2$ and BrCl surface emissions calculated from Equation 10 (purple) and the Br
atom production rate due to gas-phase radical recycling calculated from Equation 11 (orange).
Panel C:  The fraction of total Br atom production due to production from $Br_2$ and BrCl surface
emissions. In all panels, model output is smoothed by hourly averaging. The grey shaded box
represents a period of missing $Br_2$ observations. Time is expressed in Alaska Standard Time.
