# Peer review of "Bromine atom production and chain propagation during springtime Arctic ozone depletion events in Barrow, Alaska Chelsea R. Thompson, 1,a,b Paul B. Shepson, 1,2 Jin Liao, 3,c,d L. Greg Huey, 3 Chris Cantrell 4,e, Frank Flocke4, and John Orlando4 1Department of Chemist"

_Atmospheric Chemistry and Physics, 2015_

## Referee Comment (RC1) · Anonymous Referee #1 · 9 Jul 2016

[Summary]

If my guess is right, a majority of modelers who have put their hands on the development of a model of tropospheric bromine chemistry know, sort of heuristically, that the partitioning of inorganic bromine in the troposphere is highly sensitive to heterogeneous recycling (e.g., HOBr + HBr -> Br2 + H2O) in/on the aerosols (see, for example, Yang et al., 2005). So I wasn't surprised too much while reading the present paper by Thompson and co-workers despite the tone of its message. In my opinion, the main value of the present paper is the use of the "radical chain length" as an objective measure. I find it very interesting. That being said, I feel that the authors should revise their discussions and, possibly, even redo their model calculations as detailed below. One of
my major concerns is that the authors seem to have precluded much too easily a possibility for the under-representation of heterogeneous reactions in/on the aerosols while discussing the missing source of Br2 in their model calculations. I would recommend the publication of this work in ACP with major revisions.

[Specific comments]

1. For the analysis of the same dataset obtained at Barrow in the spring 2009, Liao et al. (2012) apparently used aerosol physical parameters from in-situ measurements for estimating the rate of HOBr uptake onto aerosols. On the other hand, I cannot find a description of how the present study deals with the aerosol volume and surface area and their temporal variability to constrain the rate of reactions involving aerosols. This aspect needs to be discussed thoroughly before concluding the predominant contribution of the surface snowpack in explaining the levels of Br2 measured in the field.

2. Observational data for Br2 are apparently missing for the majority of period between March 29-30 (Figure 2A; see also, Liao et al., 2012). Among the entire period of March 25-31 studied here, the period of March 29-30 is quite distinct in that ozone is recovered to near-background levels at 20-30 ppbv, which I understand is important for examining a contrast between ODE and non-ODE conditions. I suggest the authors to comment on the unavailability of Br2 data between March 29-30, how they have managed to fill in this data gap and what it means to the completeness of their model-assisted analysis. I also feel that the time series in Figures 2, 3, 5, 6, 7, 8A and 9 should be marked to indicate that the Br2 measurements were missing during March 29-30.

3. Any explanation for apparently much higher short-term variability in simulated BrO (black line) than in-situ measurements (red dots), especially during March 29-31, in Figure 2B? Although not necessarily relevant, it is during this period of time that disagreement between modeled and measured HOBr becomes really bad.

4. Lines 299-318 (Chain lengths defined by Method 1 & Method 2): I believe, "J_BrONO2 * [BrONO2]" should be taken away from the numerator of the Method 1

formula. It should also be moved from the numerator to the denominator in the Method 2 formula. According to the authors' definition, BrONO2 is the product of a termination reaction (hence k * [BrO] * [NO2] is in the denominator of the Method 1 formula). If these are not simple typographic errors, the authors should recalculate the chain lengths with corrected formulae.

5. Line 346-348: If the authors rationalize the exclusion of the photolysis of organobromine compounds from the chain initiation term (denominator) in Equation (6), there's not much point for including the OH-attack on CH3Br and CHBr3 instead. I would have liked the formula better if the authors had included all the following terms in the denominator of Equation (6): J_BrONO2 * [BrONO2], J_BrNO2 * [BrNO2], J_CHBr3 * [CHBr3] and k * [HBr] * [OH].

6. I would like to see plots equivalent to Figures 6-7 for the chain initiation. How dominant is the photolysis of Br2 in there?

7. The role of BrO + HO2 followed by the photolysis of HOBr in the net loss of ozone has been already recognized to be relatively important in the context of chemistry in the springtime polar boundary layer (Piot and von Glasow, 2008, see their introduction; Hausmann and Platt, 1994; Sander et al., 1997). In particular, Hausmann and Platt (1994, section 5 and Fig. 8) and Sander et al. (1997, section 3.1 and Fig. 2) conducted some useful calculations that can be compared with the analysis performed in the present study. Therefore, the discussion of the role of BrO + HO2 should start from Section 3.4 on ozone loss rate (rather than Section 3.5) where Figure 8 could be expanded to include additional lines and dots with contributions from k*[BrO]*[HO2] added to 2*(k*[BrO]^2 + k*[BrO]*[ClO]). And the authors should refer to those two earlier studies at first and then explain what is new and perhaps different in the present study. Then, in Section 3.5, the authors must clarify how much of HOBr thus generated ends up in heterogeneous reactions on the aerosol and snow surfaces rather than photolysis in the gas phase. Also, it seems useful to mention the dominance of Br + HCHO as a source of HO2 (Thompson et al., 2015) under the conditions of the air

studied here and what it means to the contributions of BrO + HO2 to the ozone loss and the heterogeneous formation of Br2 and BrCl in the snow and the aerosols. The role of aerosol uptake in controlling the HOBr mixing ratios has been discussed using the same dataset from Barrow by Liao et al. (2012), which should be referred to and discussed in the context of the present study.

8. Are the aqueous-phase (or surface) reactions of BrNO2 and BrONO2 with halide anions in aerosols and snow grains included in the model? I don't see them in Table 1, but they could be rather important at high levels of NOx such as on March 25.

9. Although I understand it useful to refer to a possible involvement of iodine photochemistry, there is no unequivocal evidence for its strong contributions at Barrow as much as assumed in the "High Iodine" scenario. Since, despite its hypothetical nature, this topic has been addressed once in their earlier paper (Thompson et al., 2015), I suggest the authors not to stress it much too strongly in the present paper. For example, the first paragraph of Section 3.3 may indeed need a brief comment on the hypothetical nature of the "High Iodine" scenario.

10. Line 229: Here, the first-order rate constant of transfer out of the snowpack of emitted species is referred to as the same as the first-order rate constant of transfer of depositing species from the air to the snowpack. But this seems odd, considering a large difference in assumed volumes between the air (the entire boundary layer) and the snowpack.

11. Line 402: Looking at the NOx data presented in Custard et al. (2015), NOx appears to have been pretty high on March 26 as well.

12. Lines 468-470: This finding is not new; the authors should read Barrie et al. (1988, page 140), Zeng et al. (2006, equation (2) and paragraph [15]) and Holmes et al. (2010, equation (2)).

13. Lines 496-497: Cite Bottenheim et al. (1990) along with Shepson et al. (1996)

when referring to the role of CH3CHO as a sink for Br-atom.

14. Lines 599-601: There are no actual model results presented in this paper regarding the role of snow gain acidity in the Br2 surface fluxes. This statement should be dropped; otherwise, cite the model study of Toyota et al. (2014), which have demonstrated the role of pH in the snow QLL in the production of Br2 using a condensed-phase chemical mechanism similar to the one employed in the present study.

[Technical suggestions]

1. Throughout the manuscript, the authors use the nomenclature "mole ratio" when referring to what is normally called "mixing ratio" or "mole fraction" in atmospheric chemistry. I suggest the authors to use either of the latter two.

2. Explain what the J-max is (Line 200) – a theoretical daily maximum under the clear sky? Also, how are the J-values are scaled under cloudy conditions? A sentense or two will do.

3. Line 209: . . . using this mechanism; however, . . .

4. Table 1: It appears that the mass transfer of HCl between gas and condensed phases should be included, as it is probably important for the formation of BrCl in the aerosols.

[References]

Barrie, L. A., et al.: Ozone destruction and photochemical reactions at polar sunrise in the lower Arctic troposphere, Nature, 334, 138-141, 1988.

Bottenheim, J. W., et al.: Depletion of lower tropospheric ozone during Arctic spring: the Polar Sunrise Experiment 1988, J. Geophys. Res., 95, 18555-18568, 1990.

Custard, K. D., et al.: The NOx dependence of bromine chemistry in the Arctic atmospheric boundary layer, Atmos. Chem. Phys., 15, 10799-10809, doi:10.5194/acp-15-10799-2015, 2015.

[Figure]

Hausmann, M., and Platt, U.: Spectroscopic measurement of bromine oxide and ozone in the high arctic during Polar Sunrise Experiment 1992, J. Geophys. Res., 99, 25,399-25,413, 1994.

Holmes, C. D., et al.: Global atmospheric model for mercury including oxidation by bromine atoms, Atmos. Chem. Phys., 10, 12037-12057, doi:10.5194/acp-10-12037-2010, 2010.

Liao, J., et al.: Observations of inorganic bromine (HOBr, BrO, and Br2) speciation at Barrow, Alaska, in spring 2009, J. Geophys. Res., 117, D00R16, doi:10.1029/2011JD016641, 2012.

Piot, M. and von Glasow, R.: The potential importance of frost flowers, recycling on snow, and open leads for ozone depletion events, Atmos. Chem. Phys., 8, 2437-2467, doi:10.5194/acp-8-2437-2008, 2008.

Sander, R., et al.: Modeling the chemistry of ozone, halogen compounds and hydrocarbons in the arctic troposphere during spring, Tellus Ser. B, 49, 522-532, 1997.

Thompson, C. R., et al.: Interactions of bromine, chlorine, and iodine photochemistry during ozone depletions in Barrow, Alaska, Atmos. Chem. Phys., 15, 9651-9679, doi: 10.5194/acp-15-9651-2015, 2015.

Toyota, K., et al.: Air–snowpack exchange of bromine, ozone and mercury in the springtime Arctic simulated by the 1-D model PHANTAS – Part 1: In-snow bromine activation and its impact on ozone, Atmos. Chem. Phys., 14, 4101-4133, doi:10.5194/acp-14-4101-2014, 2014.

Yang, X., et al.: Tropospheric bromine chemistry and its impacts on ozone: A model study, J. Geophys. Res., 110, D23311, doi:10.1029/2005JD006244, 2005.

Zeng, T., et al.: Halogen-driven low-altitude O3 and hydrocarbon losses in spring at northern high latitudes, J. Geophys. Res., 111, D17313, doi:10.1029/2005JD006706, 2006.

---

## Referee Comment (RC2) · Anonymous Referee #2 · 23 Jul 2016

This paper uses a 0D box model with a simplified description of heterogeneous chemistry to study the cycle of bromine during ozone depletion events and non-ozone depletion conditions in Barrow, Alaska. This paper is interesting because it proposes to use the bromine chain length to study the relative importance of gas phase cycling of bromine compared to primary emissions/heterogeneous recycle of bromine, both of which have long been thought to be important for catalytic ozone destruction by bromine in the Arctic boundary layer.

Upon a first look, the paper appears to be well thought out and presented resulting in some interesting conclusions. But, upon deeper scrutiny it is clear there are some major problems with the modeling approach that need to be addressed. In its current form,

the paper does not meet the standards for publication in ACP. The paper may eventually be considered for publication, but major modifications to the modeling approach and conclusions are required.

The main problem is that the model in its current state make it impossible to answer the questions proposed by the authors. How can the authors compare chain cycling with primary emissions, when they don't model either primary emissions or heterogeneous recycling of halogens on aerosols? In their model, Br2 is fixed to the measured concentrations (not formed by processes in the model) therefore, there is a contribution from some unknown source or sink of Br2 (and hence 2Br) at each time step of the model. Another way to think about this is to ask the question – how can the authors distinguish heterogeneous chain propagation to form Br2 from surface emissions of Br2? Using the current model setup, these cannot be distinguished making the analysis of the chain length and comparison to primary emissions rates derived incorrect.

There are further warnings of serious problems for using this modeling approach to understand how BrO + BrO competes with primary emissions of Br2 (or heterogeneous recycling of HOBr to Br2). These include:

1. The measured concentrations of BrO and HOBr are not actually captured by the model (Figure 2, 30 & 31 March). Given this, the regime where BrO + BrO dominates cannot be studied adequately.

2. The model over predicts very high HOBr concentrations (Figure 2, 30 & 31 March), leading to the conclusion that there may be something wrong with their heterogeneous uptake and recycling to Br2 of HOBr. This also limits the ability of the model to compare/contrast Br2 formed in the gas phase with Br2 formed via heterogeneous chemistry.

3. Equation 9 is extremely misleading – attributing all Br2 photolysis to surface emissions. Br2 is also the product of the BrO + BrO self-reaction. Br2 formed from BrO+BrO in the gas phase must also be considered in this equation and throughout the paper

when discussion the Br chain length.

4. There is no mention of the prior work by Toyota et al. (2014, 2011) on the importance of surface recycling for HOBr to sustain ozone depletion events/bromine release. These modeling papers should be referenced and fully discussed. References to other modeling work on this topic is also missing and should be more thoroughly discussed/thought through.

5. There is no discussion of the role of HOBr formation (XO + HO2) and subsequent photolysis in bromine recycling. This is a major problem, given its role in chain propagation.

6. Do the authors propose to separate primary emissions (bromine explosion initiation step) from emissions on snow/aerosols via HOBr recycling?

7. There is no discussion of the role of heterogeneous recycling of BrNO3. The paper cannot be considered for publication without heterogeneous recycling through BrNO3 uptake included in the model.

8. Given the high NOx levels at Barrow, a thorough discussion (including figures that demonstrate that nitrogen oxides are adequately modeled) is needed. The role of chain recycling through NOx must be considered and discussed.

9. Given the nature of surface emissions from snow, which are the main focus of the conclusions, there are likely to be vertical gradients in the concentrations of both nitrogen and bromine containing species. What are the implications of ignoring these vertical gradients on the conclusions regarding the chain length compared to primary surface emissions?

10. Are the model rate constants independent of temperature as suggested by Table 1? If there is no temperature dependence of rate constants included, the model is likely wrong during portions of the day as many of the reaction rates depend on temperature and the surface temperature at Barrow has an important diurnal cycle during

this portion of the year.

The paper cannot be published in the current format. The heterogeneous reactions involved in halogen recycling are not thoroughly discussed/considered in chain propagation. The most recent halogen review papers have more clear discussions of these radical cycles (e.g. Abbatt et al., 2012). There is also a clear discussion of these cycles in earlier reviews, for example Monks et al. (2005).

The authors have done some modeling work that can potentially be extended to answer questions about the relationship between bromine cycling and ozone depletion events. However, with their current model the authors cannot adequately compare the bromine chain length with primary emissions using a 0D chemical box model setup that relies on fixing the essential radial precursors to measurements. The authors should consider a major revision that answers a set of scientific questions that are appropriate for the model they use.

Minor comments:

1. Figure 1 is already found in Thompson 2015 (Figure 1a). Is there a specific reason for it to be included as a separate figure here? The ozone time series can simply be added to what is now Figure 2.

2. The authors should mention in their reaction schemes (equations listed) that HOBr also reacts to OH + Br in the gas phase.

References:

Abbatt, J. P. D., et al.: Halogen activation via interactions with environmental ice and snow in the polar lower troposphere and other regions, Atmos. Chem. Phys., 12, 6237-6271, doi:10.5194/acp-12-6237-2012, 2012.

Monks, P. S.: Gas-phase radical chemistry in the troposphere, Chem. Soc. Rev., 34, 376–395, 2005.

[Figure]

Toyota, K., Dastoor, A. P., and Ryzhkov, A.: Air–snowpack exchange of bromine, ozone and mercury in the springtime Arctic simulated by the 1-D model PHANTAS – Part 2: Mercury and its speciation, Atmos. Chem. Phys., 14, 4135-4167, doi:10.5194/acp-14-4135-2014, 2014.

Toyota, K., McConnell, J. C., Lupu, A., Neary, L., McLinden, C. A., Richter, A., Kwok, R., Semeniuk, K., Kaminski, J. W., Gong, S.-L., Jarosz, J., Chipperfield, M. P., and Sioris, C. E.: Analysis of reactive bromine production and ozone depletion in the Arctic boundary layer using 3-D simulations with GEM-AQ: inference from synoptic-scale patterns, Atmos. Chem. Phys., 11, 3949-3979, doi:10.5194/acp-11-3949-2011, 2011.

---

## Author Comment (AC1) · 15 Oct 2016

We would like to sincerely thank the reviewers for their time efforts in reviewing this manuscript. We believe that the suggestions offered have improved this work. Please find our responses to the comments below.

**Reviewer #1**

If my guess is right, a majority of modelers who have put their hands on the development of a model of tropospheric bromine chemistry know, sort of heuristically, that the partitioning of inorganic bromine in the troposphere is highly sensitive to heterogeneous recycling (e.g., HOBr + HBr -> Br2 + H2O) in/on the aerosols (see, for example, Yang et al., 2005). So I wasn't surprised too much while reading the present paper by Thompson and co-workers despite the tone of its message. In my opinion, the main value of the present paper is the use of the "radical chain length" as an objective measure. I find it very interesting. That being said, I feel that the authors should revise their discussions and, possibly, even redo their model calculations as detailed below. One of my major concerns is that the authors seem to have precluded much too easily a possibility for the under-representation of heterogeneous reactions in/on the aerosols while discussing the missing source of Br2 in their model calculations. I would recommend the publication of this work in ACP with major revisions.

The reviewer is correct that in our submitted manuscript we implied that the reactions of HOBr on the snowpack was the source of $Br_2$ (cf. Pratt et al., 2013; Lehrer et al., 2004), while in fact reactions on aerosols could be an important source.  In the revision, in multiple places, we now specify that it is a condensed phase process, either involving aerosols and/or the snowpack.

 [Specific comments]

1. For the analysis of the same dataset obtained at Barrow in the spring 2009, Liao et al. (2012) apparently used aerosol physical parameters from in-situ measurements for estimating the rate of HOBr uptake onto aerosols. On the other hand, I cannot find a description of how the present study deals with the aerosol volume and surface area and their temporal variability to constrain the rate of reactions involving aerosols. This aspect needs to be discussed thoroughly before concluding the predominant contribution of the surface snowpack in explaining the levels of Br2 measured in the field.

We in fact cannot quantitatively distinguish between the snowpack and aerosols because of, as we have discussed in the paper, the inability given the current state of scientific knowledge, to simulate snowpack photochemistry, on the basis of fundamentals.  Implying that we could was an error, corrected in the revised manuscript.  We only state that much or most of the Br atoms are derived from $Br_2$ photolysis, where most of the $Br_2$ is produced from the condensed phase (by inference). The revised manuscript now quantifies how much of the $Br_2$ is likely from condensed phase emissions by considering, and correcting for, the known gas-phase production pathways [Lines 712-722]. Our model was originally developed with the intention of incorporating a full heterogeneous mechanism with both the snow and aerosol parameterized separately based on the model framework of Michalowski et al. (2000). In that work, they used a constant average aerosol surface area of 3.95 x 10$^{-7}$ cm$^2$/cm$^3$ from measurements made at Alert by Staebler et al. (1994), with a maximum aerosol radius of r = 0.1 µm, and this is also what we used. Though we did not use the measurements from Barrow, the value we used is consistent with observations of aerosol surface area at Barrow, which ranged between 9 x $10^{-8}$ $cm^2/cm^3$ and 40 x $10^{-7}$ $cm^2/cm^3$ (Liao et al. 2012). We have added Lines 293-311 in the revised draft explaining this. However, because we were unable to reproduce observed $Br_2$ and $Cl_2$, we opted to forgo all of the heterogeneous production mechanisms (both aerosol and snow), and simply constrain to the observations. The "surfaces" are only important in terms of a sink, and there is no chemistry or production (with the one exception of BrCl, as noted in Lines 275-280). Because of this, we cannot, and do not, attempt to separate $Br_2$ being produced from aerosols or snow, and instead simply lump these two together as "surface." The use of the word surface perhaps was unclear, therefore we have added a sentence on Lines 307-310 to specifically state this.

2. Observational data for Br2 are apparently missing for the majority of period between March 29-30 (Figure 2A; see also, Liao et al., 2012). Among the entire period of March 25-31 studied here, the period of March 29-30 is quite distinct in that ozone is recovered to near-background levels at 20-30 ppbv, which I understand is important for examining a contrast between ODE and non-ODE conditions. I suggest the authors to comment on the unavailability of Br2 data between March 29-30, how they have managed to fill in this data gap and what it means to the completeness of their model-assisted analysis. I also feel that the time series in Figures 2, 3, 5, 6, 7, 8A and 9 should be marked to indicate that the Br2 measurements were missing during March 29-30.

We have added discussion in the Methods section (Lines 268-275 of the revised draft) addressing the missing $Br_2$ observations and explaining how this period was filled in based on the observations of the adjacent days. Our estimation of $Br_2$ during this data gap results in BrO values in good agreement with observations, therefore, we believe the estimations to be reasonable. In all of the figures in the revised manuscript presenting analyses from the model (new Figures 3,5-10), we have indicated the period of missing $Br_2$ values with a shaded box.

3. Any explanation for apparently much higher short-term variability in simulated BrO (black line) than in-situ measurements (red dots), especially during March 29-31, in Figure 2B? Although not necessarily relevant, it is during this period of time that disagreement between modeled and measured HOBr becomes really bad.

The higher short term variability is a result of reading in (unsmoothed, but sub-sampled) atmospheric datasets from a variety of instruments that all have their own inherent noise associated with the measurements, which is then compounded through multiple chemical reaction calculations. This variability is not important to the conclusions of the modeling, since we look at daily trends in the data, and it is a distraction that unnecessarily complicates the figures. Therefore, we have opted to smooth the model output using hourly averages and have implemented this in all figures. With regard to HOBr, we have revised the model so that we now constrain to the HOBr observations. This doesn't impact the conclusions of the paper since HOBr is important only as a photolytic precursor to Br atoms.

4. Lines 299-318 (Chain lengths defined by Method 1 & Method 2): I believe, "J_BrONO2 * [BrONO2]" should be taken away from the numerator of the Method 1 formula. It should also be moved from the numerator to the denominator in the Method 2 formula.
According to the authors' definition, BrONO2 is the product of a termination reaction (hence k * [BrO] * [NO2] is in the denominator of the Method 1 formula). If these are not simple typographic errors, the authors should recalculate the chain lengths with corrected formulae.

This is correct, thank you. This correction has been applied to the equations, and all values recalculated, with the figures updates accordingly.

5. Line 346-348: If the authors rationalize the exclusion of the photolysis of organobromine compounds from the chain initiation term (denominator) in Equation (6), there's not much point for including the OH-attack on CH3Br and CHBr3 instead. I would have liked the formula better if the authors had included all the following terms in the denominator of Equation (6): J_BrONO2 * [BrONO2], J_BrNO2 * [BrNO2], J_CHBr3 * [CHBr3] and k * [HBr] * [OH].

Regarding the organobromine reactions, this is a fair point. Both the OH reactions and photolysis are negligible terms. Following this suggestion, however, we have added bromoform photolysis to Equation 6, as well as $BrONO_2$ photolysis, $BrNO_2$ photolysis, and HBr + OH, and stated Lines 454-463 of the revision that these are present in the interest of completeness.

6. I would like to see plots equivalent to Figures 6-7 for the chain initiation. How dominant is the photolysis of Br2 in there?

This is a very good suggestion. We have added a figure (new Figure 6) showing the time-varying rates for the initiation reactions. In Panel A, it is clear that $Br_2$ photolysis far dominates the initiation. In Panel B, we have removed the $Br_2$ photolysis term so that the other initiations can be seen. Lines 562-588 of the revised manuscript have been added to discuss this new figure.

7. The role of BrO + HO2 followed by the photolysis of HOBr in the net loss of ozone has been already recognized to be relatively important in the context of chemistry in the springtime polar boundary layer (Piot and von Glasow, 2008, see their introduction; Hausmann and Platt, 1994; Sander et al., 1997). In particular, Hausmann and Platt (1994, section 5 and Fig. 8) and Sander et al. (1997, section 3.1 and Fig. 2) conducted some useful calculations that can be compared with the analysis performed in the present study. Therefore, the discussion of the role of BrO + HO2 should start from Section 3.4 on ozone loss rate (rather than Section 3.5) where Figure 8 could be expanded to include additional lines and dots with contributions from k*[BrO]*[HO2] added to 2*(k*[BrO]^2 + k*[BrO]*[ClO]). And the authors should refer to those two earlier studies at first and then explain what is new and perhaps different in the present study.

Our reason for using the $2(k[BrO]^2 + k[BrO][ClO])$ estimation is because this method has been used in several previous works to estimate the $O_3$ loss rate and, as such, has been accepted to be a valid estimation. We show that this method accounts for only 44% of the ozone loss rate, and that estimating $O_3$ loss with only BrO observations is inaccurate, which is one of the new outcomes of our study. To address this suggestion, however, we have now added to this figure (new Figure 9B) an estimation for $O_3$ loss using only $k[BrO][HO_2]$ (as done in Hausmann and

Platt, 1994, Figure 8) and also an estimation combining all 3 of these gas-phase cycles. Consistent with Hausmann and Platt, 1994, Figure 8, the $O_3$ loss rate estimated by $k[BrO][HO_2]$ is much slower than even $2(k[BrO]^2 + k[BrO][ClO])$ and only accounts for 18% of the $O_3$ loss rate. Combining the 3 gas-phase cycles gives an estimation that is faster, but still only accounts for 60%. We have added this discussion in the text in Section 3.4 Lines 660-675 of the revised manuscript.

In regards to Sander et al. (1997), Figure 2, we have added discussion in Section 3.5, Lines 694-707 comparing our results for the $BrO + HO_2$ reaction rate to that calculated by Sander et al. Our reaction rate is much faster than that of Sander, and we consider that this is likely due to the large differences in $HO_2$ mixing ratios between our models.

Then, in Section 3.5, the authors must clarify how much of HOBr thus generated ends up in heterogeneous reactions on the aerosol and snow surfaces rather than photolysis in the gas phase.

Because we cannot accurately simulate heterogeneous chemistry in the snowpack component of the model, and because of the very large uncertainties associated with any estimation of HOBr deposition velocities to either snow or aerosol, we could not address this using our model with any sort of accuracy. Thus, as stated above, we chose to constrain the model to HOBr observations. This, of course, makes addressing this question completely impossible. In order to address this question, then, we ran another version of our model with a variable surface deposition, based on daily wind speeds, that was tuned to achieve reasonable agreement with observations. This is discussed in Lines 355-371, and new Figure 2 has been added to show the difference in HOBr between these simulations. Using the simulation with the variable HOBr deposition, we were able to estimate that 19% of HOBr is lost to photolysis and 80% is lost to "surfaces" (some combination of aerosol and snow). The low contribution of HOBr photolysis to Br initiation is consistent with Zeng et al. (2006). This point has been added to the text in Section 3.3, Lines 579-588, where we discuss the initiation terms.

Also, it seems useful to mention the dominance of Br + HCHO as a source of HO2 (Thompson et al., 2015) under the conditions of the air studied here and what it means to the contributions of BrO + HO2 to the ozone loss and the heterogeneous formation of Br2 and BrCl in the snow and the aerosols.

We have included lines 699-701 of the revised manuscript, stating this pathway as a primary source of $HO_2$, in the discussion of the $BrO + HO_2$ reaction rate comparison with Sander et al. (1997). Even at the higher $HO_2$ levels encountered at Barrow, in comparison to the Sander et al. results, the $BrO + HO_2$ gas-phase cycle is a minor contributor to ozone depletion, and so would be even less so with lower $HO_2$. We do not simulate heterogeneous production of $Br_2$, so we cannot address what impact this would have on $Br_2$ production in our model. As a thought experiment, one could reasonably argue that to a first approximation lower $HO_2$ values would lead to lower HOBr, which would in turn decrease heterogeneous $Br_2$ production (and vice versa). This is an interesting question, but not one that we can address with our specific model.

The role of aerosol uptake in controlling the HOBr mixing ratios has been discussed using the same dataset from Barrow by Liao et al. (2012), which should be referred to and discussed in the context of the present study.

We have added further discussion of this aspect of the Liao et al. (2012) study to the Methods section in Lines 304-307, and also to the new discussion of Figure 2 (showing HOBr in different simulations) on Lines 358-363.

8. Are the aqueous-phase (or surface) reactions of BrNO2 and BrONO2 with halide anions in aerosols and snow grains included in the model? I don't see them in Table 1, but they could be rather important at high levels of NOx such as on March 25.

These aqueous phase reactions are not included because we do not produce $Br_2$ or $Cl_2$ through the heterogeneous mechanism. We do include deposition of these species to the surface as a sink term following the same method as in Custard et al., 2015 (however, we have noticed that these deposition terms were not included in Table 1, and we apologize for that oversight). Without these sink terms, the mixing ratios of $BrNO_2$ and $BrONO_2$ predicted by the model become very high, thus they are included, but no chemistry is done with them once they deposit. The photolysis of $BrNO_2$ as a bromine atom initiation source is relatively higher on March 25 (see new Figure 6), but this still does not compete with $Br_2$. As is mentioned in the Methods section, only BrCl is produced through the (admittedly simplistic) heterogeneous chemistry, and the mixing ratios of 0-10 pptv are consistent with the relatively sparse BrCl observations that we do have from Barrow. BrCl is not a significant factor in bromine initiation compared to $Br_2$.

9. Although I understand it useful to refer to a possible involvement of iodine photochemistry, there is no unequivocal evidence for its strong contributions at Barrow as much as assumed in the "High Iodine" scenario. Since, despite its hypothetical nature, this topic has been addressed once in their earlier paper (Thompson et al., 2015), I suggest the authors not to stress it much too strongly in the present paper. For example, the first paragraph of Section 3.3 may indeed need a brief comment on the hypothetical nature of the "High Iodine" scenario.

This is a good suggestion and we agree. For this paper, we have now omitted the High Iodine scenario altogether, instead using only the more realistic Low Iodine scenario. Iodine is now only addressed in Table 2 and Figure 5 in terms of chain length, and in Figure 7 in terms of potential for bromine propagation.

10. Line 229: Here, the first-order rate constant of transfer out of the snowpack of emitted species is referred to as the same as the first-order rate constant of transfer of depositing species from the air to the snowpack. But this seems odd, considering a large difference in assumed volumes between the air (the entire boundary layer) and the snowpack.

The mass transfer in and out of the snowpack was made equivalent in the same manner as Michalowski et al. (2000), which assumes that the mass transfer is limited only by the rate of vertical mixing. However, mass transfer is only important for the depositing species for surface sink terms since we do not produce $Br_2$ or $Cl_2$ heterogeneously, thus, there is in effect no mass transfer out of the snowpack.

11. Line 402: Looking at the NOx data presented in Custard et al. (2015), NOx appears to have been pretty high on March 26 as well.

There are sporadic enhancements in $NO_x$ through the late morning of March 26, but were not seen from mid-day onwards. The main difference between $NO_x$ on March 25 and March 26 is that $NO_x$ was continuously enhanced on the 25[th] with concomitant continuous enhancement of CO, whereas on the 26[th] there were only sporadic and short-lived spikes of $NO_x$ that did not display the same CO enhancements. For the NO and $NO_2$ values that were read into the model, they were subsampled at 10 minute intervals, which eliminated the short transient spikes on the morning of 26 March. Because they were transients, and not correlated with CO, we do not consider them to have a large impact on the chemistry. The figure referred to in Custard et al. (2015) was produced by smoothing all of the data by averaging, such that those transients skewed the smoothed data upwards.

12. Lines 468-470: This finding is not new; the authors should read Barrie et al. (1988, page 140), Zeng et al. (2006, equation (2) and paragraph [15]) and Holmes et al. (2010, equation (2)).

We have revised the tone of this statement and acknowledge that this has been previously recognized and applied to the Br steady state calculations presented in Zeng et al. and Holmes et al.

13. Lines 496-497: Cite Bottenheim et al. (1990) along with Shepson et al. (1996) when referring to the role of CH3CHO as a sink for Br-atom.

This has been corrected, as suggested by the reviewer.

14. Lines 599-601: There are no actual model results presented in this paper regarding the role of snow gain acidity in the Br2 surface fluxes. This statement should be dropped; otherwise, cite the model study of Toyota et al. (2014), which have demonstrated the role of pH in the snow QLL in the production of Br2 using a condensed phase chemical mechanism similar to the one employed in the present study.

We have dropped these lines from the text, and also have cited the modeling studies of Toyota et al. 2011 and 2014 in our mention of the pH of the QLL, Lines 777-781 of the revision.

[Technical suggestions]

1. Throughout the manuscript, the authors use the nomenclature "mole ratio" when referring to what is normally called "mixing ratio" or "mole fraction" in atmospheric chemistry. I suggest the authors to use either of the latter two.

We have changed all of these instances to mixing ratio.

2. Explain what the J-max is (Line 200) – a theoretical daily maximum under the clear sky? Also, how are the J-values are scaled under cloudy conditions? A sentense or two will do.

We have added lines 234-238 text stating that "All other photolysis reactions were scaled to $J(NO_2)$ in the modeling code using the maximum $J$ coefficients ($J_{max}$) for 25 March (a clear-sky day) as a scaling factor. For cloudy days, this method assumes that $J$ coefficients for other photolytically-active species are attenuated in a manner that is proportional to $J(NO_2)$."

3. Line 209: . . . using this mechanism; however, . . .

This has been corrected.

4. Table 1: It appears that the mass transfer of HCl between gas and condensed phases should be included, as it is probably important for the formation of BrCl in the aerosols.

This has been added to Table 1.

**Reviewer #2**

This paper uses a 0D box model with a simplified description of heterogeneous chemistry to study the cycle of bromine during ozone depletion events and non-ozone depletion conditions in Barrow, Alaska. This paper is interesting because it proposes to use the bromine chain length to study the relative importance of gas phase cycling of bromine compared to primary emissions/heterogeneous recycle of bromine, both of which have long been thought to be important for catalytic ozone destruction by bromine in the Arctic boundary layer. Upon a first look, the paper appears to be well thought out and presented resulting in some interesting conclusions. But, upon deeper scrutiny it is clear there are some major problems with the modeling approach that need to be addressed. In its current form, the paper does not meet the standards for publication in ACP. The paper may eventually be considered for publication, but major modifications to the modeling approach and conclusions are required.

The main problem is that the model in its current state make it impossible to answer the questions proposed by the authors. How can the authors compare chain cycling with primary emissions, when they don't model either primary emissions or heterogeneous recycling of halogens on aerosols? In their model, Br2 is fixed to the measured concentrations (not formed by processes in the model) therefore, there is a contribution from some unknown source or sink of Br2 (and hence 2Br) at each time step of the model. Another way to think about this is to ask the question – how can the authors distinguish heterogeneous chain propagation to form Br2 from surface emissions of Br2? Using the current model setup, these cannot be distinguished making the analysis of the chain length and comparison to primary emissions rates derived incorrect.

What we can do, and did explain more clearly in the revision, is calculate what fraction of Br atoms are derived from gas phase radical reactions vs. from emissions from the condensed phase. The latter can indeed be from aerosols or from the snowpack. We cannot distinguish, because as stated on pages 11 and 12 of the revision, it is not possible to accurately simulate snow phase photochemistry. It is thus true that by constraining the model to $Br_2$ observations we cannot inherently say how much of the observed $Br_2$ is from surfaces and how much is from gas-phase reactions. To address this, in Section 3.3 and the new Figure 6, where we discuss the initiation reactions, we make clear that the $Br_2$ photolysis includes all $Br_2$. In the original Section 3.5 where we had determined Br atom production from "surface emissions", we had assumed that all $Br_2$ was from the surface. We have now modified that to correct for $Br_2$ produced in the gas-phase by creating a proxy in the model, $Br_2*$, that is $Br_2$ formed only in gas-phase reactions. Thus, by subtracting $Br_2*$ from $Br_2$, we get our best estimate for $Br_2$ that is from condensed phase chemistry. This has also allowed us to determine that, on average, $Br_2$ produced via gas-phase reactions can account for 35% of the observed $Br_2$. Lines 708-722 of Section 3.5 in the revision have been added to discuss this.

It is also important to understand that the chain length refers only to gas-phase radical reactions, and by definition, does not include heterogeneous reactions, and our lack of knowledge of the details of heterogeneous chemistry has no impact on this analysis. We have attempted to make this point clearer in the text.

There are further warnings of serious problems for using this modeling approach to understand how BrO + BrO competes with primary emissions of Br2 (or heterogeneous recycling of HOBr to Br2). These include:

1. The measured concentrations of BrO and HOBr are not actually captured by the model (Figure 2, 30 & 31 March). Given this, the regime where BrO + BrO dominates cannot be studied adequately.

We have now constrained our model to the observations of HOBr, and have added a new Figure 2 to the revision. The new BrO values are shown in Figure 1. Constraining HOBr slightly decreased BrO. The variability of the modeled output has also now been smoothed with hourly averaging (see response to Reviewer 1, comment 3). With the constrained HOBr, the model now explicitly represents HOBr observations, and we feel that the modeled BrO is also in good agreement with the observations.

2. The model over predicts very high HOBr concentrations (Figure 2, 30 & 31 March), leading to the conclusion that there may be something wrong with their heterogeneous uptake and recycling to Br2 of HOBr. This also limits the ability of the model to compare/contrast Br2 formed in the gas phase with Br2 formed via heterogeneous chemistry.

As discussed, we chose to constrain the model to HOBr observations, since we cannot simulate the snowpack photochemistry. We did, however, also run a version of the model with variable HOBr deposition, based on daily wind speeds, that was tuned to better reproduce the observations. This is shown in new Figure 2C in the revision. This version of the model was used to determine that 19% of HOBr is lost to photolysis and 80% is lost to surfaces (some combination of snow and aerosol). All other calculations (e.g. the chain length, and relative sources of Br atoms) were redone using the model constrained to HOBr. Because we also constrain to $Br_2$, our model does not inherently separate $Br_2$ from surface emissions and $Br_2$ from gas-phase reactions. For this reason, we incorporated our gas-phase $Br_2$ proxy, $Br_2^*$, as discussed above.

3. Equation 9 is extremely misleading – attributing all Br2 photolysis to surface emissions. Br2 is also the product of the BrO + BrO self-reaction. Br2 formed from BrO+BrO in the gas phase must also be considered in this equation and throughout the paper when discussion the Br chain length.

We agree that assuming all $Br_2$ is from the surface (as was done in Equation 9) is inaccurate. Therefore, we have now corrected that equation (new Equation 10), by subtracting $Br_2$ formed via gas-phase reactions ($Br_2^*$) from the total observed $Br_2$. Thus, new Equation 10, now represents our best estimate for $Br_2$ from surface emissions from the condensed phase (aerosol + snowpack). By doing this, we were also able to determine that gas-phase production of $Br_2$ accounts for 35% on average of the observed $Br_2$. The source of the $Br_2$ has no impact on the chain length calculation, however, as the photolysis of $Br_2$ (regardless of source) is an initiation term for Br. That said, in discussion of the new Figure 6, where we show the rates of the initiation terms, we do now point out clearly that the $Br_2$ photolysis term includes all $Br_2$ regardless of source [Lines568-570].

4. There is no mention of the prior work by Toyota et al. (2014, 2011) on the importance of surface recycling for HOBr to sustain ozone depletion events/bromine release. These modeling papers should be referenced and fully discussed. References to other modeling work on this topic is also missing and should be more thoroughly discussed/thought through.

References to Toyota et al. (2011, 2014) have been added to both the Methods and the Results sections.

5. There is no discussion of the role of HOBr formation (XO + HO2) and subsequent photolysis in bromine recycling. This is a major problem, given its role in chain propagation.

This comment was also made by Reviewer 1. We have addressed this in several ways following the suggestions by Reviewer 1. We have added a new Figure 6, which shows the bromine initiation terms and demonstrates the minor role of HOBr photolysis for bromine initiation. Here, we also now discuss that we have found that photolysis accounts for 19% of the HOBr loss and surface deposition accounts for 80%, based upon our model simulation with variable and enhanced HOBr deposition (Figure 2C). This minor role of HOBr photolysis is consistent with the work of Zeng et al. (2006). Additionally, in discussing the $O_3$ loss rates in Section 3.4, we have added $O_3$ loss rate estimations using $k[BrO][HO_2]$, comparing to a similar calculation by Hausmann and Platt (1994), further confirming the minor contribution of this gas-phase pathway.

6. Do the authors propose to separate primary emissions (bromine explosion initiation step) from emissions on snow/aerosols via HOBr recycling?

We do not propose to separate $Br_2$ that is emitted from the surface in the initial bromide activation step (the mechanism for which is not fully understood, though several theories do exist) versus $Br_2$ that is emitted from the surface following HOBr recycling. Indeed, this work does not provide any information about the chemical mechanism that produces $Br_2$ from the condensed phases. In referencing "primary emissions", we were simply referring to any $Br_2$ that is emitted from a surface through any mechanism. Equation 9 (new Equation 10) has been updated and recalculated to correct for $Br_2$ formed through gas-phase pathways (such as one channel of the BrO + BrO reaction), so that the result of this equation is our best estimate for Br atoms derived from $Br_2$ that was produced in a condensed phase. Here, "condensed phase" refers to some combination of aerosol and snow.

7. There is no discussion of the role of heterogeneous recycling of BrNO3. The paper cannot be considered for publication without heterogeneous recycling through BrNO3 uptake included in the model.

As we have discussed, we are not simulating heterogeneous chemistry. We have expanded the discussion in the Methods section to make this clearer and to justify why we chose not to produce these compounds heterogeneously, citing the works of Domine et al. 2013 and Cao et al., 2014, which clearly discuss the inherent difficulties in attempting to model snow surface chemistry. Indeed, these papers argue that snow surface chemistry cannot be modeled accurately at this time given the state of our knowledge. Deposition of $BrONO_2$ (and $BrNO_2$) is included as a sink term in the same manner as in Custard et al. (2015). We regret that these deposition terms were overlooked in Table 1 and that has been corrected. As discussed, the chain length refers to gas phase radical chain propagation only, and therefore it is important to understand that explicit heterogeneous reactions have no bearing on this metric.

8. Given the high NOx levels at Barrow, a thorough discussion (including figures that demonstrate that nitrogen oxides are adequately modeled) is needed. The role of chain recycling through NOx must be considered and discussed.

The impact of $NO_x$ on the bromine chain length is discussed in Lines 487-506, and is discussed in much greater detail in Custard et al. (2015). Custard et al. also address the role of $NO_x$ in moderating the rate of ozone depletion. Because this is the main topic of that work, we don't feel that we need to expand further in this current manuscript and restate the conclusions of Custard et al. To the extent that $NO_x$ impacts recycling via $BrNO_2$ and $BrONO_2$, again, we don't/can't simulate that chemistry in the snowpack. Our model is constrained to NO and $NO_2$ observations and this figure is shown in Thompson et al. (2015).

9. Given the nature of surface emissions from snow, which are the main focus of the conclusions, there are likely to be vertical gradients in the concentrations of both nitrogen and bromine containing species. What are the implications of ignoring these vertical gradients on the conclusions regarding the chain length compared to primary surface emissions?

This is a very good point, and we have added Lines 762-777 in the Conclusions section addressing this. The lines we have added state:

"We find that between 30 – 90% of Br atoms are produced from surface emissions of $Br_2$ and BrCl, though we cannot distinguish snow sources from aerosol sources using our model. However, it is important to note that we do not know how much of the condensed phase $Br_2$ production derives from reaction R7, or from some other condensed phase process, e.g. oxidation of $Br^-$ by OH radicals (Abbatt et al., 2010). The in situ snow chamber experiments by Pratt et al. (2013) demonstrate a strong $Br_2$ source from the snowpack; similar field observations proving significant $Br_2$ emissions from Arctic aerosol are currently lacking. If the snow surface is the primary sources of these emissions, then a strong vertical gradient would be expected in the near surface boundary layer, and our estimations for the Br chain length would be only valid for the height of our measurements (~ 1 m above the snow). Strong deposition to the snow would also induce a vertical gradient in these species. If, however, aerosols are an important source of $Br_2$ (or other halogen precursors), then $Br_2$ production should occur throughout the entire height of the boundary with no significant vertical gradient, in a similar fashion as has been observed for $ClNO_2$, which is a known product of aerosol chemistry (Young et al., 2012). It is clear that vertically-resolved measurements of these halogen precursors are imperative for our understanding of halogen production in the Arctic."

10. Are the model rate constants independent of temperature as suggested by Table 1? If there is no temperature dependence of rate constants included, the model is likely wrong during portions of the day as many of the reaction rates depend on temperature and the surface temperature at Barrow has an important diurnal cycle during this portion of the year.

Most of the rate constants involved in radical recycling are indeed not very temperature dependent. We feel that this is justified in this case because ambient temperature in Barrow for the week of 25 March 2009 varied by less than 10 K between the maximum and minimum recorded daily temperatures. Also, the radical oxidation and radical-radical reactions that are of primary importance here do not have a large dependence on temperature (Atkinson et al., 2006, 2007); for example, a variability of 10 K imposes an ~1% change on the rate of ethane oxidation by Cl atoms and a <4% change on the rate of the BrO + BrO radical self-reaction. Furthermore, and as mentioned previously, the major stable chemical species driving the model, especially those VOCs that can exhibit a more important temperature dependence, are highly constrained to observations and are not allowed to freely evolve. We have added text to the Methods section Lines 202-213 addressing this point.

The paper cannot be published in the current format. The heterogeneous reactions involved in halogen recycling are not thoroughly discussed/considered in chain propagation. The most recent halogen review papers have more clear discussions of these radical cycles (e.g. Abbatt et al., 2012). There is also a clear discussion of these cycles in earlier reviews, for example Monks et al. (2005).

The chain length, by definition, is a metric for the radical chain propagation in the gas phase only, and as such, does not include heterogeneous reactions. As an example, the work by Monks et al. (2005), which we do cite in reference to the chain length, also uses the chain length metric to discuss the gas-phase cycling of HOx. The implication of a short gas-phase chain length is that bromine recycling through the heterogeneous reactions is highly important, but again, this cannot be accurately simulated on the basis of fundamentals. While previous modeling studies do show that bromine production cannot be sustained without heterogeneous chemistry, we are using a specific metric, the gas-phase chain length, to support the conclusions of those works and argue that the rate of ozone depletion cannot be accurately estimated with gas-phase reactions alone, as has been done in several papers.

The authors have done some modeling work that can potentially be extended to answer questions about the relationship between bromine cycling and ozone depletion events. However, with their current model the authors cannot adequately compare the bromine chain length with primary emissions using a 0D chemical box model setup that relies on fixing the essential radical precursors to measurements. The authors should consider a major revision that answers a set of scientific questions that are appropriate for the model they use.

Minor comments:

1. Figure 1 is already found in Thompson 2015 (Figure 1a). Is there a specific reason for it to be included as a separate figure here? The ozone time series can simply be added to what is now Figure 2.

Figure 1 and 2 have been combined. The ozone time series is now shown as Panel A in Figure 1.

2. The authors should mention in their reaction schemes (equations listed) that HOBr also reacts to OH + Br in the gas phase.

The photolysis of HOBr to OH + Br is listed in Table 1, as is the reaction of HOBr with OH.

[revised manuscript text omitted]